# Structure–function analysis of *Lactiplantibacillus plantarum* DltE reveals D-alanylated lipoteichoic acids as direct cues supporting *Drosophila* juvenile growth

Nikos Nikolopoulos[1†], Renata C Matos[2†], Stephanie Ravaud[1†], Pascal Courtin[3†], Houssam Akherraz[2], Simon Palussiere[3], Virginie Gueguen-Chaignon[4], Marie Salomon-Mallet[3], Alain Guillot[3], Yann Guerardel[5,6], Marie-Pierre Chapot-Chartier[3]*, Christophe Grangeasse[1]*, François Leulier[2]*

[1]Molecular Microbiology and Structural Biochemistry, CNRS UMR 5086, Université Claude Bernard Lyon 1, Lyon, France; [2]Institut de Génomique Fonctionnelle de Lyon, Ecole Normale Supérieure de Lyon, CNRS UMR 5242, Université Claude Bernard Lyon 1, Lyon, France; [3]Université Paris-Saclay, INRAE, AgroParisTech, Micalis Institute, Jouy-en-Josas, France; [4]Protein Science Facility, CNRS UAR3444, INSERM US8, Université Claude Bernard Lyon 1, Ecole Normale Supérieur de Lyon, Lyon, France; [5]Institute for Glyco-core Research (iGCORE), Gifu University, Gifu, Japan; [6]Univ. Lille, CNRS, UMR 8576 - UGSF - Unité de Glycobiologie Structurale et Fonctionnelle, Lille, France

*For correspondence:
marie-pierre.chapot-chartier@
inrae.fr (M-PierreC-C);
christophe.grangeasse@ibcp.
fr (CG);
francois.leulier@ens-lyon.fr (FL)

†These authors contributed
equally to this work

Competing interest: The authors
declare that no competing
interests exist.

Reviewing Editor: Karina B
Xavier, Instituto Gulbenkian de
Ciência, Portugal

**Abstract** Metazoans establish mutually beneficial interactions with their resident microorganisms. However, our understanding of the microbial cues contributing to host physiology remains elusive. Previously, we identified a bacterial machinery encoded by the *dlt* operon involved in *Drosophila melanogaster*'s juvenile growth promotion by *Lactiplantibacillus plantarum*. Here, using crystallography combined with biochemical and cellular approaches, we investigate the physiological role of an uncharacterized protein (DltE) encoded by this operon. We show that lipoteichoic acids (LTAs) but not wall teichoic acids are D-alanylated in *Lactiplantibacillus plantarum*[NC8] cell envelope and demonstrate that DltE is a D-Ala carboxyesterase removing D-Ala from LTA. Using the mutualistic association of *L. plantarum*[NC8] and *Drosophila melanogaster* as a symbiosis model, we establish that D-alanylated LTAs (D-Ala-LTAs) are direct cues supporting intestinal peptidase expression and juvenile growth in *Drosophila*. Our results pave the way to probing the contribution of D-Ala-LTAs to host physiology in other symbiotic models.

## Editor's evaluation

This is an important study on the role of a bacterial cell wall component, D-alanylated lipoteichoic acid, as a cue in *Drosophila melanogaster*-microbiome interactions. Overall, the evidence presented to support the conclusions is compelling. The approach combines crystallography with biochemical and cellular assays that take advantage of both fly and bacterial mutants to demonstrate a physiological role in juvenile growth promotion. The work will be of broad interest to those studying host-microbe interactions, particularly aspects related to immunology and metabolism, mediated by the microbiome.

## Introduction

Metazoans establish mutually beneficial interactions with their resident microorganisms (*McFall-Ngai et al., 2013*). These interactions contribute to different aspects of host physiology, including juvenile growth, a postnatal developmental process marked by rapid body-size increase and organ maturation (*Schwarzer et al., 2018*). Juvenile growth results from the integration of environmental cues with the organism's intrinsic genetic potential, driven by energy and nutritional demands. Harsh environmental conditions, notably nutrient deprivation, result in linear and ponderal growth failure (i.e. stunting) (*Nabwera et al., 2022*), as well as alteration in gut microbiota maturation (*Subramanian et al., 2014*), which can be mitigated by microbial interventions and/or microbiota-directed nutritional interventions (*Barratt et al., 2022*). Despite some recent advances, the understanding of how gut microbes buffer the deleterious effect of undernutrition and contribute to healthy juvenile growth remains elusive.

*Drosophila melanogaster* (referred here as *Drosophila*) is a valuable experimental model to study the physiological consequences and underlying mechanisms of host–commensal bacteria interactions (*Douglas, 2018*; *Grenier and Leulier, 2020*). Bacterial strains associated with *Drosophila* influence multiple physiological processes, including juvenile growth, and on several occasions, the underlying symbiotic cues, that is, bacterial molecules directly impacting host functionalities, have been identified. For instance, amino acids produced by symbiotic bacteria can inhibit the production of the neuropeptide CNMamide in the gut, which shape food foraging behavior by repressing preference for amino acids (*Kim et al., 2021*). Bacterial metabolite such as acetate produced by strains of *Acetobacter pomorum* is necessary to support juvenile growth (*Shin et al., 2011*) by altering the epigenome of enteroendocrine cells and stimulating the secretion of the intestinal hormone Tachykinin (*Jugder et al., 2021*; *Kamareddine et al., 2018*). In the context of microbe-mediated *Drosophila* juvenile growth promotion (*Storelli et al., 2011*), peptidoglycan (PG) fragments from *Lactiplantibacillus plantatum* (*Lp*) cell walls are directly sensed by peptidoglycan recognition receptors (PGRPs) in *Drosophila* enterocytes. This recognition signal, via the IMD/NF-kappaB pathway, promotes the production of intestinal peptidases, which helps juveniles optimizing the assimilation of dietary proteins to support their systemic growth (*Erkosar et al., 2015*).

Recently, in an effort to further characterize the bacterial machinery involved in *Lp*-mediated juvenile growth promotion, we identified through forward genetic screening the *pbpX2-dltXABCD* operon as an important determinant of *Lp*-induced *Drosophila* larval growth (*Matos et al., 2017*). The first gene of the operon, *pbpX2* (here renamed *dltE* for D-Ala-LTA Esterase, see below), is uncharacterized and annotated as a serine-type D-Ala-D-Ala-carboxypeptidase putatively involved in PG maturation cleaving the terminal D-alanine (D-Ala) residue of the peptide stem in newly made muropeptides. Remarkably, *Lp* PG precursors do not contain a terminal D-Ala in peptide stems but a terminal D-lactate (D-Lac) (*Ferain et al., 1996*) raising the question of the biochemical activity of DltE. On the other hand, the remaining *dltABCD* encode a multi-protein machinery responsible for the D-alanylation of teichoic acids (TA) in diverse Gram-positive bacteria (*Perego et al., 1995*) and in *Lp* (*Nikolopoulos et al., 2022*). TAs are anionic polymers localized within the Gram-positive bacteria cell wall, representing up to 50% of the cell envelope dry weight and present in two forms: wall teichoic acids (WTAs), which are covalently bound to PG, and lipoteichoic acids (LTAs), which are anchored to the cytoplasmic membrane (*Rohde, 2019*). Study of an isogenic mutant of *Lp* NC8 (*Lp^{NC8}*) strain, deleted for the entire *dlt* operon, revealed that this operon is indeed essential to D-Ala esterification to the cell envelope (most likely on TAs) and that purified D-alanylated cell envelopes triggered intestinal peptidase expression and support *Drosophila* juvenile growth (*Matos et al., 2017*). These results suggested that TA modifications are important cues shaping commensal-host molecular dialog. However, which type of TAs and whether their modification is directly involved remains unaddressed and an indirect influence of the D-alanylation process on PG maturation and PG sensing by the host could not be excluded.

To address these standing questions, we investigate here the structure, the biochemical activity, and the physiological role of DltE. We also study the impact of DltE on TA structure and modifications in *Lp* cell envelope and test the relative contribution of purified PG, WTA, and D-alanylated-LTA (D-Ala-LTA) from *Lp* cell envelope to support *Drosophila* growth. Our results establish that DltE is not a carboxypeptidase modifying *Lp* PG but rather a D-Ala esterase acting upon D-Ala-LTA. After characterizing the chemical structure of LTAs and WTAs, we show that only LTAs but not WTAs are

D-alanylated in $Lp^{NC8}$ cell envelopes and we demonstrate that D-Ala-LTAs, in addition to PG, are direct cues supporting intestinal peptidase expression and juvenile growth in *Drosophila*.

## Results

### Structure of the extracellular domain of DltE

DltE is annotated as a putative serine type D-Ala-D-Ala carboxypeptidase. However, sequence comparison with canonical members of this protein family revealed low sequence identity (*Figure 1—figure supplement 1*). Additionally, while the catalytic S-X-X-K motif (Motif 1) is conserved in DltE, the second (Y-X-S) and third (K/H-T/S-G) motifs that complete the active site are altered with only the catalytic Tyr and the Gly residues being conserved, respectively. This suggests a modified active site environment with potentially different catalytic and/or substrate binding mechanisms. Supporting this, we showed that the extracellular catalytic domain of DltE (DltE$_{extra}$ 34–397) is not able to bind penicillin although this property is a hallmark of serine type D-alanyl-D-alanine carboxypeptidases that belongs to the PBP (penicillin binding proteins) protein family (*Figure 1—figure supplement 2*).

These features prompted us to determine the structure of the extracellular domain of DltE (DltE$_{extra}$) by X-ray crystallography (*Supplementary file 1*). The domain consists of two subdomains, an α-β sandwich (residues 76–126 and 246–397) that form a five strand (β1–β5) antiparallel β-sheet flanked by 5 α-helices (α2, α9–α12) and an α-helix-rich region (residues 127–245) folding in 6 α-helices (α3–α8) (*Figure 1a* and *Figure 1—figure supplement 1*). DltE is structurally homologous to the D-Ala-D-Ala carboxypeptidase R61 (DDCP) from *Streptomyces* (*Figure 1b and c*; *Kelly et al., 1986*; *McDonough et al., 2002*) that exhibits the classical β-lactamase fold of PBPs (*Figure 1b*). The positions of the active site S-X-X-K motif ($^{128}$SIQK$^{131}$), containing the catalytic Ser128 and Lys131 residues, and the catalytic Tyr213 from the second motif are conserved (*Figure 1a and b* and *Figure 2*). These last two residues are expected to function as a general base during the acylation step and function as a relay mechanism for the transfer of a proton from the incoming substrate to the departing catalytic serine. In contrast, several striking features distinguish DltE from DDCP and more generally from PBPs and DD-carboxypeptidases. The β3-strand that usually bears the third motif and defines one side of the catalytic cavity is not conserved in terms of length and position (*McDonough et al., 2002*; *Figure 1c*). Second, DltE contains an extra α-helix (α1) at the N-terminus of the catalytic domain (*Figure 1c*), as well as three major structural differences affecting the regions lining the sides of the active site of DDCP and responsible for its cavity-like shape (*Figure 1c, e and g*): (1) a 15-amino acid long loop connecting α4 and α5 in DltE (residues 175–190 named Loop I on *Figure 1c and d* and *Figure 1—figure supplement 1*) that replaces a longer region of 20 residues that is folded in α-helices in DDCP; (2) the mostly unstructured segment ranging from residues 328–346 in DltE and located between α11 and β3 (Loop II on *Figure 1c and d* and *Figure 1—figure supplement 1*) is 10 residues longer in DDCP in which it forms an antiparallel β-sheet of three small strands that follows the main 5-stranded β-sheet; and (3) the loop between α8 and α9 (residues 257–283 in DltE) connecting the two subdomains is slightly shorter in DltE and adopts a different position (*Figure 1c*). This last difference is of particular interest because this loop was described as the Ω-like loop in β-lactamases and PBP proteins (*Fetrow, 1995*; *Yi et al., 2016*) and shown to play a key role in the maintenance of the active site topology and in the enzymatic activity (*Figure 1e* and *Figure 1—figure supplement 1*). The global architecture of the active site cavity of DltE is therefore reshaped and different from that of DDCP enzymes, adopting a deeper cleft topology largely opened on both sides of the protein (*Figure 1d–g*). Together with its inability to bind penicillin, such structural features strongly suggest that DltE possesses an alternative substrate recognition mode and/or a different substrate specificity than PBPs.

### DltE is not active on the PG stem depsipeptide

Besides the apo form structure, we obtained two other DltE structures in complex with tartrate or TCEP (Tris(2-carboxyethyl)phosphine hydrochloride), two compounds arising from the crystallization conditions (*Supplementary file 1* and *Figure 2a and b*). Intriguingly, these two molecules are anchored by an interaction network that involves eight different residues, among which are the conserved active site nucleophile Ser128 and the base Tyr213 (Ser 62 and Tyr 159 in DDCP; *Figure 2a, b and e*). Importantly, the catalytic Ser128 with the hydroxyl group of its side chain positioned at only 2.7 Å away from a carboxylic group of TCEP and tartrate is ideally placed as it would activate a substrate

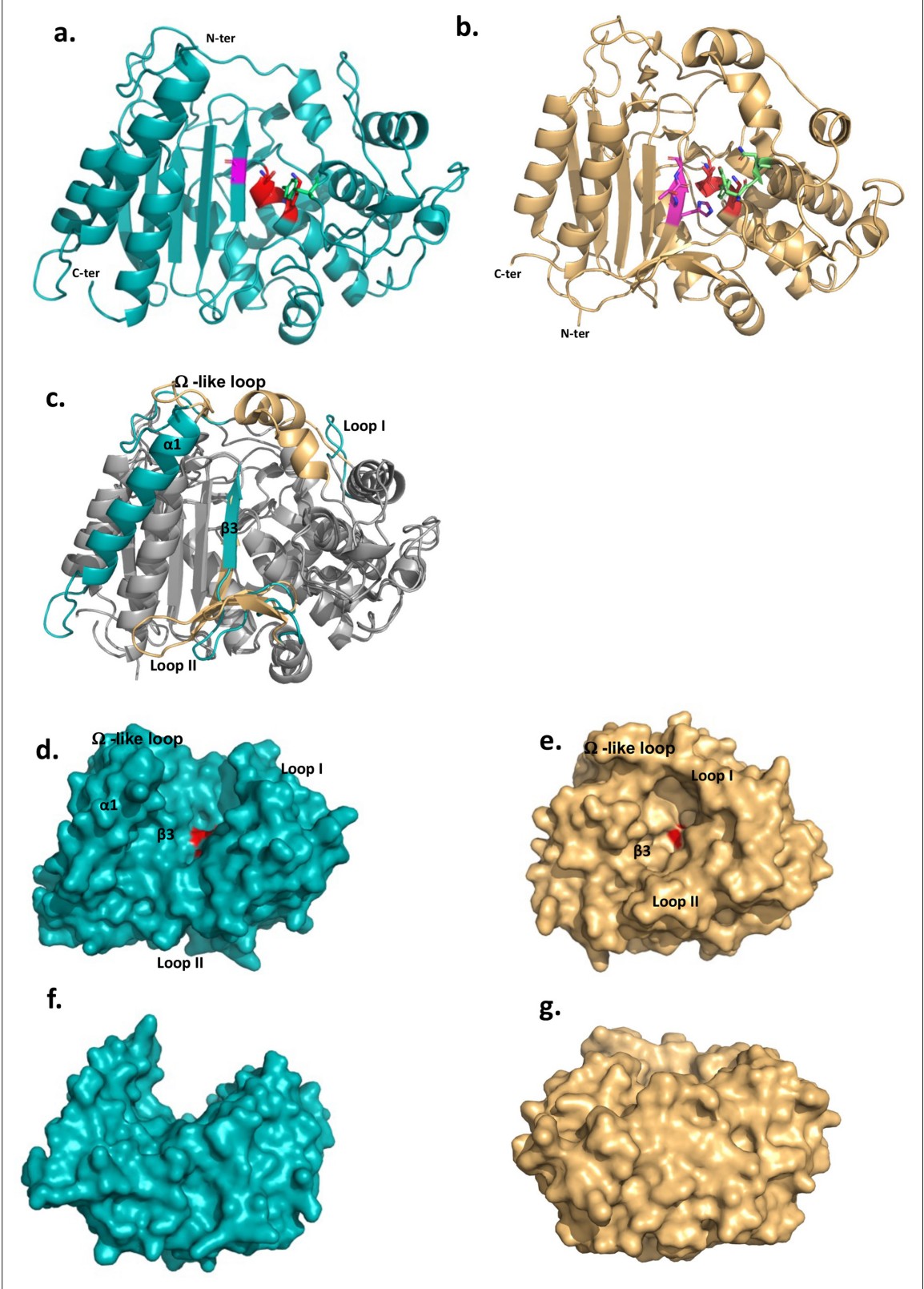

**Figure 1.** The 3D structures of *L. plantarum* DltE~extra~ and its structural comparison with the *Streptomyces* R61 D-Ala-D-Ala carboxypeptidase structure. (**a**) Cartoon representations of the 3D X-ray structure of *L. plantarum* DltE~extra~ and (**b**) of the canonical D-Ala-D-Ala carboxypeptidase from *Streptomyces* sp. R61 (DDCP) (PDB ID 1HVB/1IKG). The N- and C-termini are indicated. The canonical DDCP conserved motifs 1 (SXXK, with S the catalytic Ser), 2 (YXN), and 3 ((K/H)(S/T)G) are highlighted in red, green, and pink, respectively. The first motif is strictly conserved in DltE ($^{128}$SIQK$^{131}$) and located as for DDCP

*Figure 1 continued on next page*

*Figure 1 continued*

at the beginning of the α-helix-rich region (α3 helix in DltE, α2 in DDCP) at the interface with the β-sheet. Only the Tyr and Gly residues are conserved in the motifs 2 and 3, respectively. The position of the catalytic Tyr213 from the second motif is also conserved and found in the loop connecting α5 and α6, close to the catalytic dyad. (**c**) Superimposition between DltE$_{extra}$ and DDCP (PDB ID 1HVB/1IKG) 3D structures. The common structural cores that exhibit a classical β-lactamase fold or penicillin binding (PB) fold are colored in gray. They are well superimposed with rms deviation calculated at 2.2 Å on around 300 residues. The main structural differences are highlighted in teal for DltE and wheat for DDCP and indicated on the figure: the N-terminal α1 helix of DltE, the β3-strand that contains the motif 3, the Loop I, the Loop II, and the Ω-like loop. (**d**, **f**) and (**e**, **g**) Surface representation of DltE$_{extra}$ and DDCP. (**d**, **e**) and (**f**, **g**) are shown in the same orientation. (**f**) and (**g**) are rotated by 90° along a horizontal axis compared to (**d**) and (**e**), respectively. The catalytic Ser, colored in red, is buried at the bottom of the catalytic cavity in DDCP and lying in the middle of a large cleft in DltE. The structural elements that define the active site architecture and differ between DltE and DDCP are indicated.

The online version of this article includes the following source data and figure supplement(s) for figure 1:

**Source data 1.** DltE 3D structure.

**Source data 2.** Raw SDS-PAGE analysis of the purity of DltE$_{extra}$.

**Source data 3.** Labeled SDS-PAGE analysis of the purity of DltE$_{extra}$.

**Source data 4.** Size-exclusion chromatography (SEC) of the Ni-affinity purified DltE$_{extra}$ on a Superdex 200 10/300 GL.

**Source data 5.** Microscale thermophoresis (MST) normalized dose–response data for the binding interaction between penicillin and *S. pneumoniae* PBP2b.

**Source data 6.** Microscale thermophoresis (MST) normalized dose–response data for the binding interaction between penicillin and DltE$_{extra}$.

**Figure supplement 1.** Sequence comparison of DltE with the *Streptomyces* R61 D-Ala-D-Ala carboxypeptidase (DDCP).

**Figure supplement 2.** Production of DltE$_{extra}$ used for structure determination and biochemical assays.

for the enzyme acylation event as described for DDCP (*McDonough et al., 2002*). This suggests that the tartrate and TCEP molecule may mimic the nature and position of the natural substrate of DltE in its active site and that the acylation/deacylation step of the catalytic reaction is probably preserved in DltE (*Figure 2b and c*). In addition, tartrate or TCEP established a series of interactions with the structural elements specific to DltE (see above) and not found in DDCP (*Figure 2e*). Notably, a Tyr residue (Tyr338 in DltE) interacting with TCEP or tartrate replaces the conserved arginine (Arg285 in DDCP) in carboxypeptidases. This arginine is key for the recognition of the terminal carboxylate of the peptide substrate and the carboxypeptidase activity (*McDonough et al., 2002*; *Figure 2b, d and e*). Furthermore, the interactions involving motif 2 and 3 that stabilizes the substrate in DDCP and in particular the penultimate D-Ala residue of the peptide substrate are also largely altered (*Figure 2b, d and e*). In DltE, the hydrophobic contact with TCEP or tartrate is only maintained with the conserved Tyr213 and Gly348 of motif 2 and motif 3, respectively. Instead, residues specific to DltE (Tyr338, His347, Arg345, and Leu349) stabilize the TCEP or tartrate molecules (*Figure 2a and b*). Taken collectively, our structural data suggest that DltE would not be able to recognize the D-Ala-D-Ala terminal end of a stem peptide as DDCP does and that DltE would not be a DD-carboxypeptidase.

In *L. plantarum* PG precursors, a D-Lac residue rather than the more common D-Ala is present in position 5 of the peptide stem with an ester bond between D-Ala[4] and D-Lac[5] (*Ferain et al., 1996*). Tartrate and TCEP molecules having chemical groups that are also found in D-Lac (*Figure 3—figure supplement 1*), we hypothesized that the DltE structural features described above could reflect its ability to act as a carboxylesterase trimming the terminal D-Lac in mature PG. However, two enzymes, namely, DacA1 and DacA2, homologous to the D-Ala-D-Ala-carboxypeptidase DacA (PBP3) of *Streptococcus pneumoniae* (*Morlot et al., 2005*) could be potentially responsible for the D-Lac hydrolysis. Thus, we generated an $Lp^{NC8}$ mutant deficient for *dacA1* and *dacA2* and analyzed the PG composition. As shown in *Figure 3—figure supplement 2a*, we detected the presence of disaccharide-depsipentapeptide (GlcNAc-MurNAc-L-Ala-D-Gln-mDAP-D-Ala-D-Lac). As D-Lac residues are not detected in WT strain (*Bernard et al., 2011*), this indicates that DacA1 and/or DacA2 could behave as carboxylesterases removing the terminal D-Lac. To confirm this, we purified the disaccharide-depsipentapeptide from the Δ*dacA1*Δ*dacA2* strain and measured the ability of the purified recombinant catalytic domains of DacA1 and DltE to catalyze D-Lac hydrolysis. We observed that DltE was not able to cleave the ester bond between D-Ala[4] and D-Lac[5] from the disaccharide-depsipentapeptide, whereas purified DacA1 efficiently did (*Figure 3*). These results indicate that DltE is not a carboxylesterase active on PG stem depsipentapeptide.

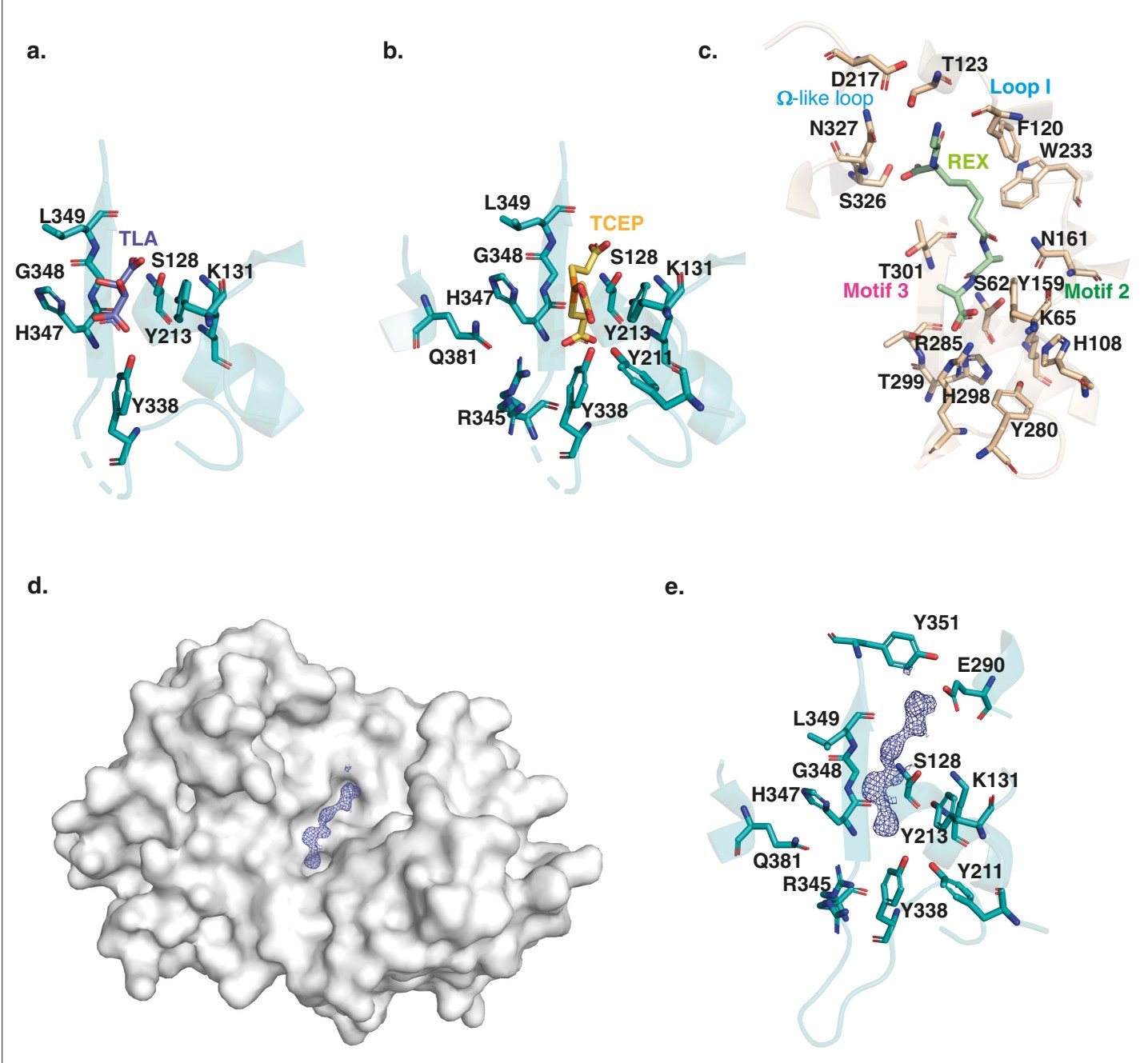

**Figure 2.** The substrate binding cleft of *L. plantarum* DltE. Close-up on the ligand binding site of DltE$_{extra}$ crystallized in complex with a tartare (TLA) (**a**), or TCEP (Tris(2-carboxyethyl)phosphine hydrochloride) (**b**), molecule. The residues involved in the interactions are shown as sticks. The catalytic Ser128, with the hydroxyl group of its chain side is positioned at only 2.7 Å away from a carboxylic group of the ligand is itself hydrogen bonded to the catalytic Lys131 of the motif 1 and to Tyr213, the only conserved residue of the motif 2. The three residues of the β3-strand ($^{347}$H-G-L$^{349}$) including Gly348 from motif 3 delineate one side of the active site. Tyr338 establishes a strong, almost covalent, interaction with ligand carboxyl group that is also bound to Ser128. Three additional interactions specific to DltE are observed in the TCEP-bond structure and involved Tyr211, Arg345, and Gln381. (**c**) Close-up on the ligand binding site of DDCCP in complex with fragment of the cell wall precursor (REX – glycyl-L-alpha-amino-epsilon-pimelyl-D-Ala-D-Ala) (PDB ID 1IKG). The residues involved in the interactions are shown as sticks. The catalytic Ser62 and Lys65 from motif 1 and Tyr159 from motif 2 are located in the vicinity of the last D-Ala moiety in positions similar to those observed in DltE. The rest of the substrate binding site differs significantly. The Arg285 conserved in DDCP and responsible for the carboxypeptidase activity, recognizes the terminal carboxylate of the substrate. The interactions involving motif 2 and 3 stabilize the penultimate D-Ala residue. The hydrophobic subsite composed of Trp233 and Phe120 recognizes the aliphatic portion of the peptide and is also notably absent in DltE. Finally, H-bond interactions are established with the N-terminus of the peptide substrate at the level of the loop I and Ω-like-loop. (**d**) Surface representation of DltE structure obtained from crystal soaked with LTA molecules. A long electron density compatible

*Figure 2 continued on next page*

*Figure 2 continued*

with a 2-mer polyglycerol phosphate was observed lying in the catalytic cleft. (**e**) Close-up on the interaction network around the backbone of the ligand modeled in the catalytic cleft. The interaction network that stabilizes the first half of the ligand is the same as the one described with the TCEP molecules. The second half of the electron density extends further on the catalytic cleft of DltE in the vicinity of Glu290 (in helix α9) and Tyr351 (between β3 and β4) that are not conserved in canonical DDCP.

The online version of this article includes the following source data for figure 2:

**Source data 1.** DltE 3D structure in complex with tartare (TLA) or TCEP (Tris(2-carboxyethyl)phosphine hydrochloride).

## DltE is a D-Ala esterase acting on D-alanylated-LTA

Since DltE is not active on PG and because *dltE* gene is part of the *dlt* operon in *Lp*[NC8], we hypothesized that DltE could have a D-Ala esterase activity on D-alanylated teichoic acids. To assess whether DltE could act on WTA and/or LTA, we first characterized their structure by NMR in *Lp*[NC8] and determined their alanylation levels. Multidimensional NMR spectroscopy analysis revealed that WTA from *Lp*[NC8] contained two main repetition units composed of ribitol (Rbo)-phosphate chains substituted by two α-Glc residues either in C-2 and C-4 positions (major unit) or in C-3 and C-4 positions (minor unit) (*Figure 3—figure supplement 2b*). Remarkably, no Ala substituents were detected. WTAs purified from Δ*dltXABCD* were structurally identical from those purified from *Lp*[NC8], which supports that the DltXABCD complex is not involved in the D-alanylation of WTA. In contrast to WTA, NMR analysis established that LTA from *Lp*[NC8] is constituted by repetitive units of glycerol (Gro)-phosphate chains that were either unsubstituted (Unit A) or substituted with either Ala (Unit B), α-Glc (Unit C) or Ala-6-αGlc (Unit D) groups at C-2 position of Gro residues (*Figure 3b*). Relative quantification of NMR signals associated with individual units showed that Unit A was the major form with A/B/C/D ratio of 62:19:15:4. NMR analysis of LTA extracted from Δ*dltXABCD* mutant confirmed the absence of Ala-substituted LTA, as demonstrated by the disappearance of GroB-2 and GroB1,3 signals, GlcD-6 signals and Ala-2 signals on the $^1$H-$^{13}$C HSQC spectrum and of GroB1 to 3 signals on the $^1$H-$^1$H COSY spectrum (*Figure 3—figure supplement 3*). Relative quantification of signals demonstrated that LTA from Δ*dltXABCD* mutant consisted of a simple mixture of Unit A and C in a 70:30 ratio. Taken together, these results establish that *Lp*[NC8] cell envelope carries LTAs with a distinctive pattern of structure and substitution and that only LTAs are D-alanylated. In addition, these observations confirm that the Dlt machinery is necessary to the D-alanylation of *Lp* LTAs.

Given that only LTAs are alanylated in *Lp*[NC8], we started testing DltE ability to bind LTA through microscale thermophoresis (MST) experiments. As shown in *Figure 3c*, DltE$_{extra}$ was able to efficiently bind LTA. Then, we examined the activity of DltE on D-Ala-LTA purified from *Lp*[NC8]. To achieve this, we analyzed the relative amounts of Ala esterified to Gro (Gro-Ala) and free Ala released from LTA upon incubation with or without DltE$_{extra}$. Both compounds were detected in the control sample as a result of spontaneous release of D-Ala from LTA when the test was performed without DltE (*Figure 3d* and *Figure 3—figure supplement 5*). Upon incubation of LTA with DltE, we observed a clear decrease (56% decrease) of Gro-Ala and a concomitant increase of free D-Ala. This effect was enhanced (64% decrease) with a double amount of DltE$_{extra}$ in the test. In contrast, no difference with the control sample was observed for LTA incubated with DltE$^{S128A}$ catalytic mutant protein or with DacA1 (*Figure 3d* and *Figure 3—figure supplement 4*). These results establish that DltE possesses a D-Ala esterase activity that cleaves the ester bond between the substituting D-Ala and Gro in LTA chains.

To gain structural insights into this activity, we soaked DltE crystals with LTA extracted from the Δ*dltXABCD* mutant and determined the 3D structure of such enzyme-substrate complexes (*Supplementary file 1*). We observe that the cleft harbors an additional, elongated density compatible with the presence of a 2-mer unit of the LTA molecule (*Figure 2c*). Half of the electron density is located at the same place as the tartrate and TCEP ligands (*Figure 2*) and would correspond to one unit of Gro-phosphate. It is stabilized by the same interaction network described above for tartrate and TCEP that share similar chemical groups, that is, hydroxyl, carboxyl and phosphate with LTA (*Figure 2d*, *Figure 3—figure supplement 1*). The second half of the electron density extends further on the catalytic cleft of DltE in the vicinity of Glu290 (in helix α9) and Tyr351 (between β3 and β4), forming thus a negatively charged subsite. The latter is not conserved in DDCP and replaced by a hydrophobic subsite that was proposed to recognize the aliphatic portion of the peptide (*Figure 3—figure supplement 6a and b*). These structural features are in agreement with the ability of DltE to bind

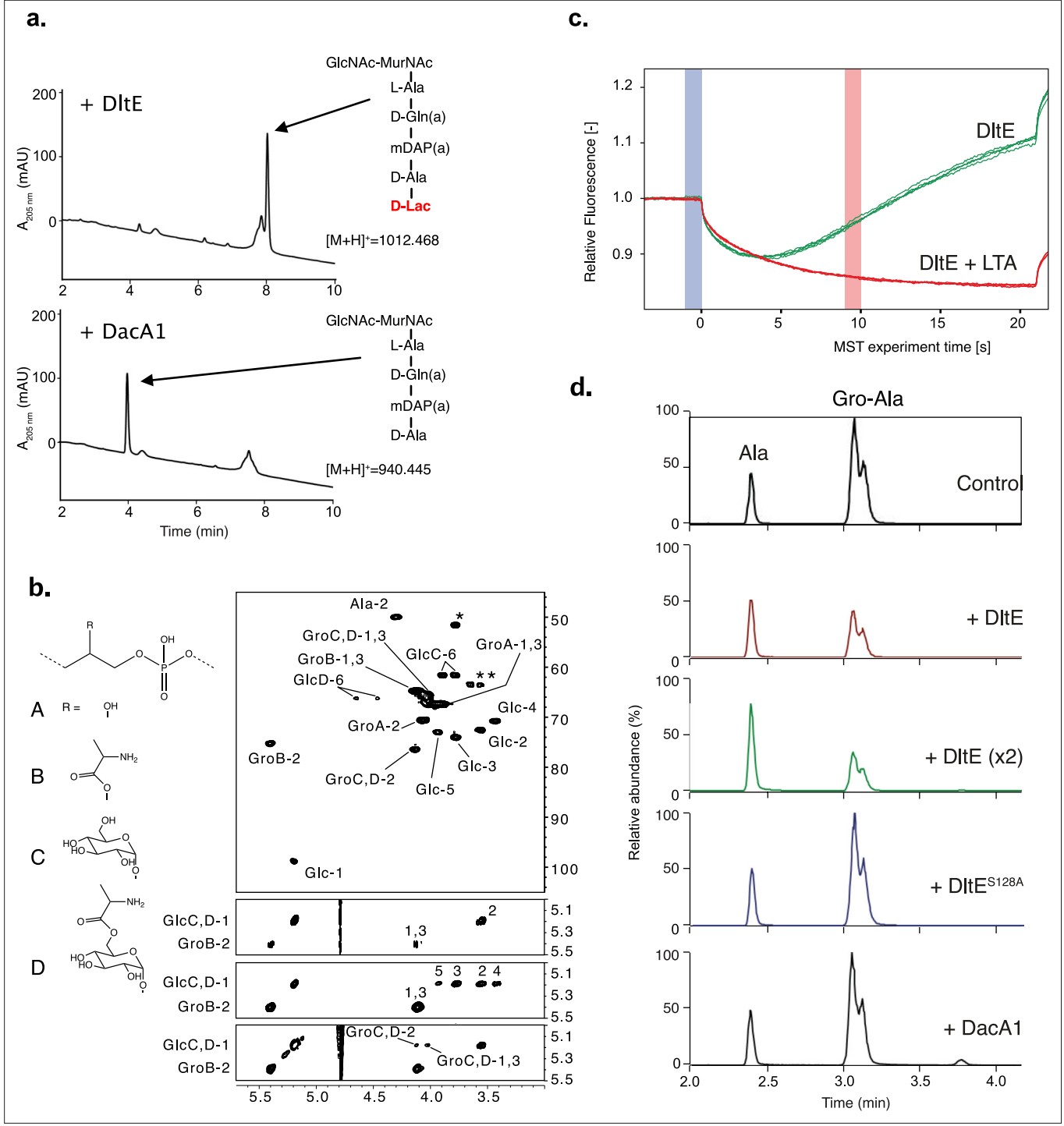

**Figure 3.** DltE is not active on peptidoglycan (PG) stem peptide but has D-Ala esterase activity on lipoteichoic acid (LTA). (**a**) Test of carboxylesterase activity of DltE on disaccharide-depsipentapeptide substrate. Purified enzymes (DltE$_{extra}$ or DacA1) were incubated with purified muropeptide, and the mixture was analyzed by ultra-high-pressure liquid chromatography (UHPLC). DacA1 taken as a control is able to release terminal D-Lac, generating disaccharide-tetrapeptide. Muropeptides were identified by mass spectrometry. (**b**) Multidimensional NMR analysis of LTA isolated from WT *L. plantarum* established the presence of four major repeating units made of phospho-glycerol (Gro) differently substituted at C-2 position by -OH group (**A**), Ala- residue (**B**), αGlc residues (**C**), or Ala-6-αGlc group (**D**). Individual spin systems of Glc residues C and D, Gro associated to A-D and Ala associated to GroB and GlcD were established from ${}^{1}$H-${}^{13}$C HSQC (top), ${}^{1}$H-${}^{1}$H COSY (second from top), and ${}^{1}$H-${}^{1}$H TOCSY (third from top) spectra in agreement with literature (***Sánchez Carballo et al., 2010***). ${}^{1}$H and ${}^{13}$C chemical shifts are reported in ***Supplementary file 2***. Ala residue was typified according to the ${}^{1}$H/${}^{13}$C chemical shifts of C1 at δ -/172.5, C2 at δ 4.29/49.9, and C3 at δ 1.63/16.36, out of which only Ala-2 is visible on the presented region of ${}^{1}$H-${}^{13}$C HSQC spectrum and Ala-1 was identified on ${}^{1}$H-${}^{13}$C HMBC spectrum (not shown). Substitution of Gro by Ala in C-2 position is

*Figure 3 continued on next page*

*Figure 3 continued*

established owing to the very unshielded GroB-2 signal at δ 5.40/75.3. Substitution of GroC,D in C-2 position was established owing to the deshielding of GroC,D-2 $^{13}$C at δ 76.4 compared to unsubstituted GroA-2 $^{13}$C at δ 70.6 and the $^1$H-$^1$H NOESY cross signal between GlcC,D-1 and GroC,D-2 (bottom spectrum). Finally, substitution of GlcD in C-6 position is observable on the $^1$H-$^{13}$C HSQC spectrum by the strong deshielding of GlcD-6 at δ 4.45–4.63/66.2 compared to unsubstituted GlcC-6 at δ 3.77–3.88/61.5[56]. Such LTAs have never been identified in *L. plantarum*, but very similar repetition units were previously characterized in LTA and wall teichoic acid (WTA) isolated from *Lactobacillus brevis* (*Sánchez Carballo et al., 2010*). However, in contrast to *L. brevis* in which WTA and LTA showed very similar Gro-based sequences, WTA and LTA from *L. plantarum* showed very different structures with repetition units based either on Rbo-phosphate for the former and on Gro-phosphate for the latter. (c) Microscale thermophoresis (MST) traces for DltE$_{extra}$ without and with purified LTA from $Lp^{NC8}$ added at a concentration of 500 µM. Relative fluorescence change reveals binding of LTA to DltE$_{extra}$ (Fhot: fluorescence at the region defined as hot 10 s after IR laser heating, Fcold: fluorescence at the region defined as cold at 0 s). (d) Test of D-Ala esterase activity of DltE on LTA from $Lp^{NC8}$. Purified LTA (100 µg) was incubated with purified enzymes: DltE$_{extra}$ (100 µg or 200 µg [×2]), DltE$_{extra}$$^{S128A}$ (100 µg) or DacA1 (100 µg). Control corresponds to LTA incubated in the same conditions without enzyme. After LTA depolymerization by HF treatment, hydrolysis products were analyzed by LC-MS/MS. Ala and Gro-Ala were identified by their m/z values (90.05 and 164.09, respectively) and MS-MS spectra (*Figure 3—figure supplement 5*). Gro-Ala was detected as a double peak, corresponding possibly to the migration of D-Ala from C2 to C1 of Gro (*Morath et al., 2001*). A chromatogram of the combined extracted ion chromatograms of each target ion species was generated for each condition. Similar results were obtained in two independent experiments.

The online version of this article includes the following source data and figure supplement(s) for figure 3:

**Source data 1.** Microscale thermophoresis (MST) data for the binding of DltE$_{extra}$ with purified lipoteichoic acid (LTA) from $Lp^{NC8}$.

**Source data 2.** DltE has D-Ala esterase activity on lipoteichoic acid (LTA).

**Figure supplement 1.** Chemical structures of (a) tartrate, (b) TCEP Tris(2-carboxyethyl)phosphine hydrochloride, (c) D-Lac, and (d) subunit of *Lp* lipoteichoic acid (LTA) molecule without any substitution.

**Figure supplement 2.** Purification of the disaccharide-depsipentapeptide from Lp NC8 dacA1dacA2 mutant and determination of Lp NC8 WTA chemical structure by NMR.

**Figure supplement 3.** Lipoteichoic acid (LTA) extracted from Δ*dltXABCD* mutant is devoid of Ala.

**Figure supplement 4.** Experimental replicates from *Figure 3d*.

**Figure supplement 4—source data 1.** Identification of Ala and Gro-Ala by MS and MS/MS.

**Figure supplement 5.** MS and MS/MS specta of Ala and Gro-Ala.

**Figure supplement 6.** Molecular surfaces of DltE$_{extra}$ (a) and DDCP (b), colored according to the electrostatic potential.

and hydrolyse D-Ala-LTA, and, together with the biochemical data, they confirm that DltE is a D-Ala esterase acting on D-Ala-LTA.

## The D-Ala esterase activity of DltE contributes to D-alanylation of the cell envelope and is required to sustain *Drosophila* juvenile growth

We previously reported that the machinery encoded by the *dlt* operon (including *dltE*) is necessary to support D-Ala esterification in *Lp* cell envelope and sustain *Drosophila* juvenile growth (*Matos et al., 2017*). As previously observed upon mild alkaline hydrolysis, D-Ala was released in appreciable amounts from $Lp^{NC8}$ (WT) cells[14] (*Figure 4a*), whereas no D-Ala was released from an isogenic mutant deleted for the entire *dlt* operon (*Matos et al., 2017*). However, we observed a significant reduction (around 70%) of D-Ala esterified to LTAs in Δ*dltE* mutant cells and an even higher reduction (around 93%) for *DltE*$^{S128A}$ catalytic mutant cells. These results indicate that the esterase activity of DltE, in addition to the activities of DltX, A, B, C, and D, contributes to the D-alanylation machinery of LTAs in bacterial cells.

We next wondered whether an active DltE protein is necessary for *L. plantarum* support to *Drosophila* juvenile growth. To this end, we compared juvenile growth (larval size at day 6 after egg laying) and developmental timing (i.e. day of 50% population entry to metamorphosis) of germ-free animals and ex-germ-free animals associated at the end of embryogenesis with either WT or Δ*dltE* and *dltE*$^{S128A}$ mutants. Both mutations did not affect the ability of *Lp* to thrive in the fly niche but significantly altered the ability of *Lp* to support larval growth and developmental timing (*Figure 4b–d* and *Figure 4—figure supplement 1*). These results demonstrate that DltE activity is required to sustain *Drosophila* juvenile growth and together with the structural and biochemical insights on DltE activity point to the importance of D-Ala-LTAs as cues supporting *Drosophila* growth.

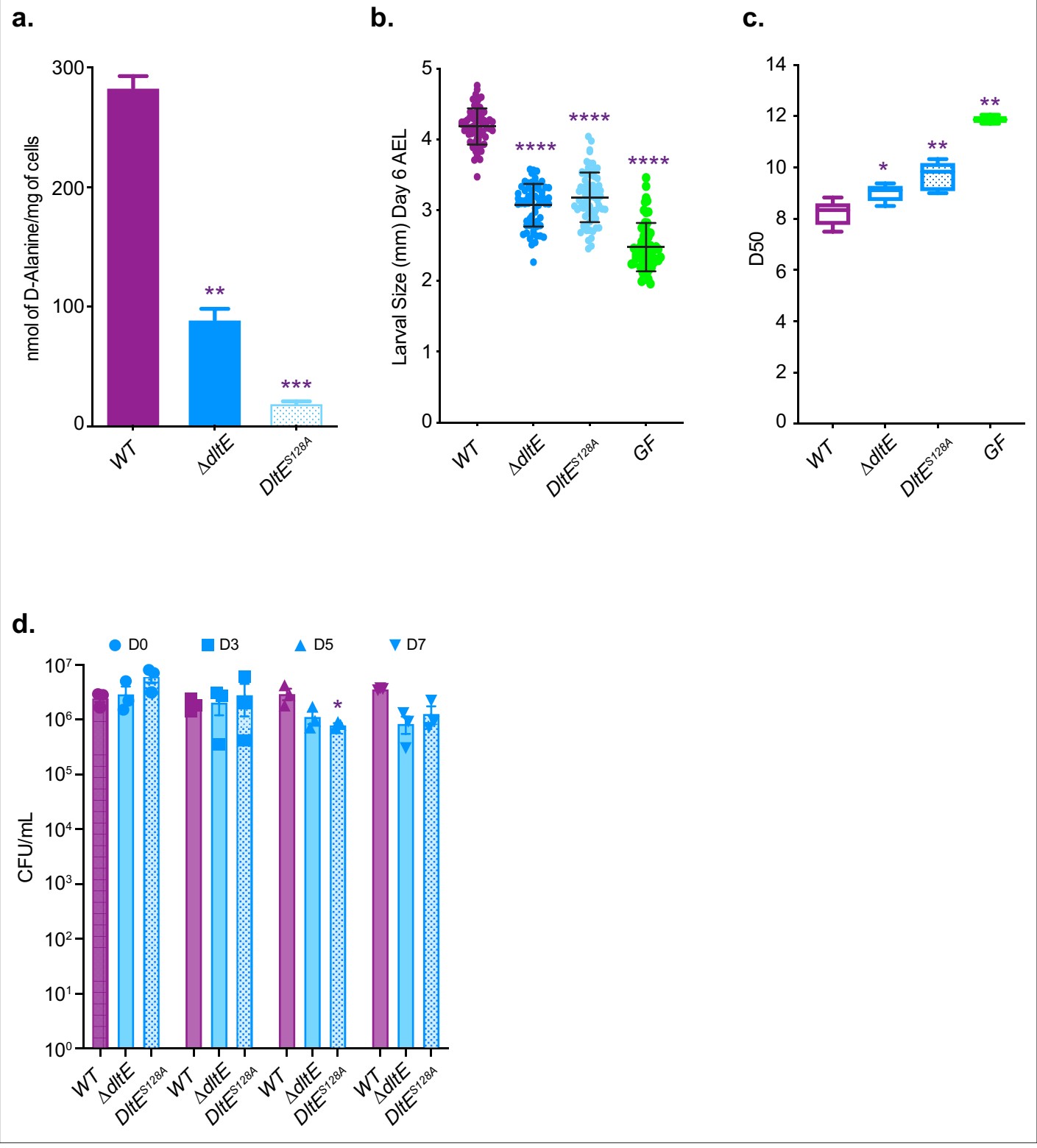

**Figure 4.** The D-Ala esterase activity of DltE contributes to D-alanylation of the cell envelope and is required to sustain *Drosophila* juvenile growth. (**a**) Amount of D-Ala released from whole cells of WT, Δ*dltE,* and *DltE^{S128A}* by alkaline hydrolysis and quantified by HPLC. Mean values were obtained from three independent cultures with two injections for each. Purple asterisks illustrate statistically significant difference with D-Ala release from WT. ***0.0001<p<0.001; **0.001<p<0.01. (**b**) Larval longitudinal length after inoculation with WT, Δ*dltE,* and *DltE^{S128A}* strains and PBS (for the germ-free [GF] condition). Larvae were collected 6 d after association and measured as described in the 'Materials and methods' section. Purple asterisks

*Figure 4 continued on next page*

*Figure 4 continued*

illustrate statistically significant difference with larval size of WT; ****p<0.0001. Center values in the graph represent means and error bars represent SD. Representative graph from one out of three independent experiments. (**c**) Time to pupation: day when 50% of pupae emerge during a developmental experiment (D50) for GF eggs associated with strains WT, Δ*dltE,* and DltE$^{S128A}$ or PBS (for the GF condition). Center values in the graph represent means. Purple asterisks illustrate statistically significant difference with WT larval size; **0.001<p<0.01; *p<0.05. (**d**) Abundance of colony-forming units (CFUs) on fly food and larvae at days 3, 5, and 7 after inoculation with WT, Δ*dltE,* and DltE$^{S128A}$. Purple asterisks illustrate statistically significant difference with WT within each day. *p<0.05.

The online version of this article includes the following source data and figure supplement(s) for figure 4:

**Source data 1.** The D-Ala esterase activity of DltE contributes to D-alanylation of the cell envelope and is required to sustain *Drosophila* juvenile growth.

**Source data 2.** The D-Ala esterase activity of DltE is required to sustain *Drosophila* juvenile growth – replicates.

**Figure supplement 1.** The D-Ala esterase activity of DltE is required to sustain *Drosophila* juvenile growth.

## D-Ala-LTAs are necessary bacterial cues supporting *Drosophila* intestinal response and juvenile growth

In order to evaluate the importance of D-Ala-LTA for *Drosophila* juvenile growth, we undertook a genetic approach by generating *L. plantarum* strains either deprived of LTA through the deletion of *ltaS* encoding for the LTA synthetase (Δ*ltaS*) (***Gründling and Schneewind, 2007***) or WTA through the deletion of *tagO* encoding for the enzyme catalyzing the first step of WTA biosynthesis (Δ*tagO*) (***Andre et al., 2011***). A similar amount of D-Ala esterified to the cell envelope was measured in Δ*tagO* mutant cells compared to WT cells, whereas a strong reduction of D-Ala (around 72%) was observed in Δ*ltaS* mutant cells (***Figure 5—figure supplement 1a***). The absence of LTA chain synthesis in Δ*ltaS* mutant was confirmed by western blot with anti-LTA monoclonal antibody (***Figure 5—figure supplement 1b***). These results confirm that LTA are D-alanylated but not WTA as determined above by NMR analysis of purified polymers (***Figure 3***). Of note, we hypothesize that the residual D-Ala released for Δ*ltaS* cells might be attached to a glycolipid precursor of LTA as previously observed in *Listeria monocytogenes* (***Webb et al., 2009***).

We then tested the respective TA defective strains for their ability to support *Drosophila* juvenile growth (***Figure 5a*** and ***Figure 5—figure supplement 2a***). When associated with germ-free animals, WT and Δ*tagO* strains support optimal *Drosophila* juvenile growth while Δ*dlt*$_{op}$ or Δ*ltaS* strains largely fail to do so despite colonizing well their host's niche (***Figure 5a***, ***Figure 5—figure supplement 1c***, and ***Figure 5—figure supplement 2a***). These data demonstrate that LTA are important molecules mediating *Lp*'s support to *Drosophila* juvenile growth. Accordingly, presence of D-Ala esterified to LTA chains (in WT and Δ*tagO* strains) correlates well with strains ability to support *Drosophila*'s growth. Indeed, D-Ala-LTA presence is required for *Lp*-mediated *Drosophila* juvenile growth promotion phenotype (***Figure 5a***, ***Figure 5—figure supplement 1a and b***, and ***Figure 5—figure supplement 2a***). In the absence of D-Ala-LTA (in Δ*dlt*$_{op}$) or LTA (in Δ*ltaS*), *Drosophila* larvae are smaller (***Figure 5a*** and ***Figure 5—figure supplement 2a***). Taken collectively these data reinforce the notion that D-Ala-LTAs from *Lp* play a key role in supporting *Drosophila* juvenile growth.

We had previously established that upon association with *Lp*, enterocytes from ex-germ-free *Drosophila* sense and signal the presence of *Lp* cells by at least two independent molecular mechanisms: (1) PGRP-LE-mediated mDAP-PG fragment recognition triggering Imd/Dredd signaling (***Erkosar et al., 2015***) and (2) sensing of bacterial cell envelope bearing D-alanylated teichoic acids and signaling by unknown host mechanisms (***Figure 5b***; ***Matos et al., 2017***). Both signals were reported to be important for maximal intestinal peptidase expression and optimal support to growth promotion by *Lp* (***Figure 5a, c and d*** and ***Figure 5—figure supplement 2a–c***; ***Matos et al., 2017***). To probe the respective contribution of LTA and WTA to this model, we tested the ability of Δ*ltaS* and Δ*tagO* strains (respectively deprived of LTA or WTA) to support *Drosophila* growth and intestinal peptidase expression in a *Drosophila* genetic background (*Dredd* mutants flies) (***Leulier et al., 2000***) that blunts the host response to mDAP-PG fragments (***Leulier et al., 2003***), allowing us to focus on the contribution of the D-alanylation signal to *Lp*-mediated growth support and intestinal peptidase induction. As previously observed upon association with the Δ*dlt*$_{op}$ mutant, *Dredd* mutant larvae associated with Δ*ltaS* are compromised in their juvenile growth potential (***Figure 5a*** and ***Figure 5—figure supplement 2a***) and their ability to stimulate intestinal peptidase expression (*Jon66Cii* and *Jon65Ai;*

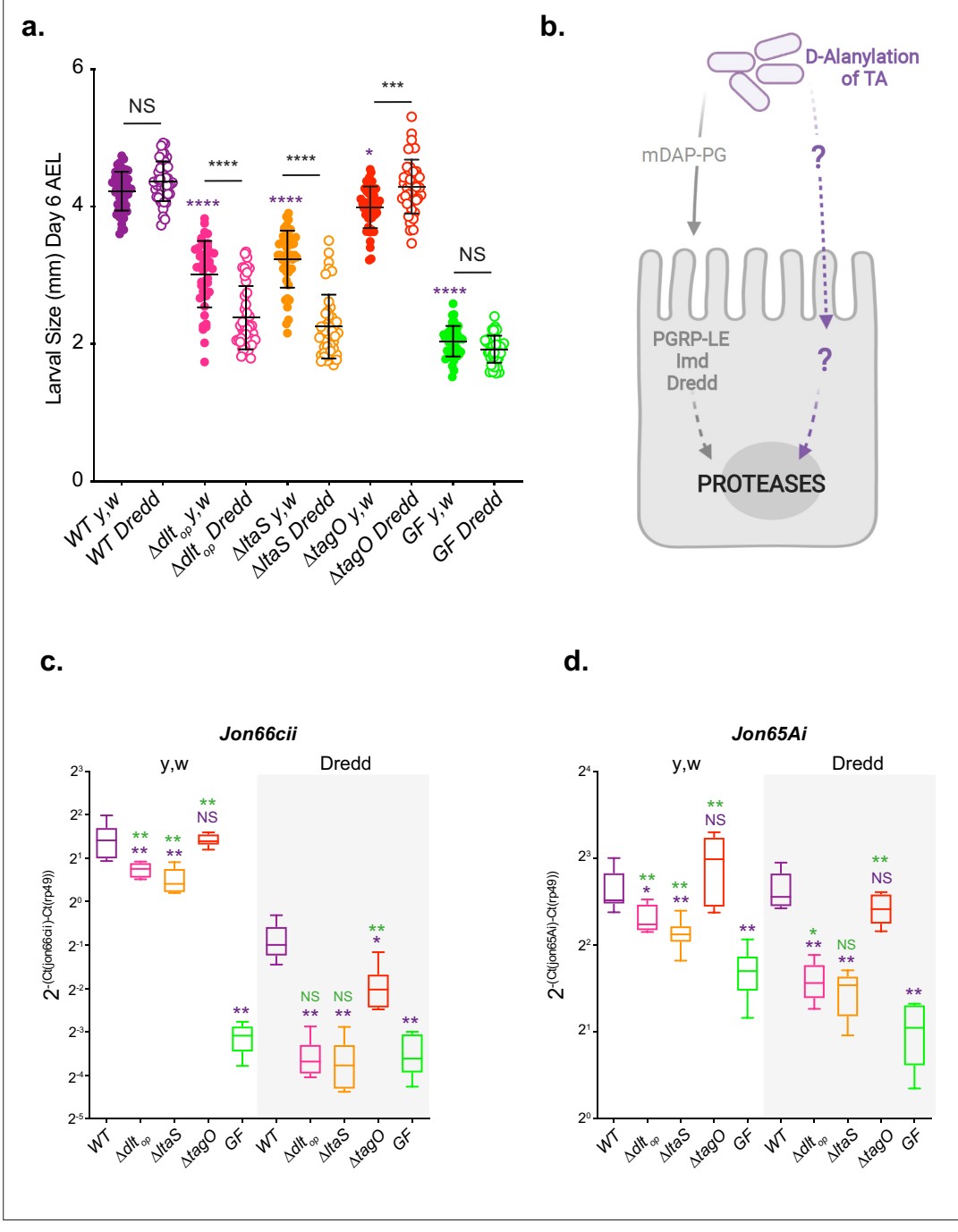

**Figure 5.** D-Ala-LTAs are necessary bacterial cues supporting *Drosophila* intestinal response and juvenile growth. (**a**) *y,w* and *y,w,Dredd* larvae longitudinal length after inoculation with $10^8$ colony-forming units (CFUs) of WT, $\Delta dlt_{op}$, $\Delta ltaS$, and PBS, for the germ-free condition. Larvae were collected 6 d after association and measured as described in the 'Materials and methods' section. The purple asterisks represent a statistically significant difference compared with WT larval size. The bars in the graph represent means and SD. NS represents the absence of a statistically significant difference; ****$p<0.000$; ***$0.0001<p<0.001$; *$p<0.05$. Representative graph from one out of three independent experiments. (**b**) Working model for *Lp* detection in *Drosophila* enterocytes: *Drosophila* sense and signal the presence of Lp cells through: (1) PGRP-LE-mediated mDAP-PG fragment recognition triggering Imd/Dredd signaling and (2) sensing of bacterial cell envelope bearing D-alanylated teichoic acids and signaling by unknown host mechanisms. Both signals were reported to be important for maximal intestinal peptidase expression. (**c**, **d**) Mean ± SD of $2^{-\Delta Ctgene/\Delta Ctrp49}$ ratios for *Jon66Cii* and *Jon65Ai* detected in dissected guts of *y,w* and *y,w,Dredd* associated with WT, $\Delta dlt_{op}$, $\Delta ltaS$ or the GF condition from five biological replicates. Representative

*Figure 5 continued on next page*

*Figure 5 continued*

graphs from one out of three independent experiments are shown. The purple asterisks represent a statistically significant difference compared with $Lp^{NC8}$ proteases expression. The green asterisks represent a statistically significant difference compared with the GF condition. NS represents the absence of a statistically significant difference compared to the GF condition or $Lp^{NC8}$. **0.001<p<0.01; *p<0.05.

The online version of this article includes the following source data and figure supplement(s) for figure 5:

**Source data 1.** D-Ala-LTAs are necessary bacterial cues supporting *Drosophila* intestinal response and juvenile growth.

**Source data 2.** D-Ala quantification and persistence on fly niche of WT, $\Delta dlt_{op}$, $\Delta ltaS$, and $\Delta tagO$ strains.

**Source data 3.** D-Ala-LTAs are necessary bacterial cues supporting *Drosophila* intestinal response and juvenile growth – replicates.

**Source data 4.** Replicate western blot detection of lipoteichoic acid (LTA) in wild-type $Lp^{NC8}$ and mutant derivatives.

**Figure supplement 1.** D-Ala content and colony forming units of strains used on *Figure 5*.

**Figure supplement 2.** D-Ala-LTAs are necessary bacterial cues supporting *Drosophila* intestinal response and juvenile growth.

---

*Figure 5c and d* and *Figure 5—figure supplement 2b and c*). However, juvenile growth and intestinal peptidase expression were not markedly affected in $\Delta tagO$-associated *Dredd* larvae (*Figure 5a–d* and *Figure 5—figure supplement 2a–c*). Importantly, we observed a cumulative effect of not having D-Ala-LTA ($\Delta dlt_{op}$) or not having LTA at all ($\Delta ltaS$) and altering host mDAP-PG sensing (compare WT -*yw*- and *Dredd* conditions, *Figure 5a–d* and *Figure 5—figure supplement 2a–c*). These results therefore support the notion that both signals are required for optimal support to larval growth by *Lp* and that in addition to mDAP-PG fragments, D-Ala-LTAs are likely an additional cue sensed by *Drosophila* enterocytes.

## D-Ala-LTAs are direct bacterial cues supporting *Drosophila* intestinal response and juvenile growth

Next, we tested the hypothesis that D-Ala-LTAs act as direct signals sensed by *Drosophila* enterocytes. To this end, we purified the major components from the cell envelope of WT and $\Delta dltXABCD$ strains: WTA, LTA, and PG, and tested their ability to rescue $\Delta dlt_{op}$ and $\Delta ltaS$-mediated larval phenotypes (*Figure 6a–e* and *Figure 6—figure supplement 1a–d*). First, germ-free, $\Delta dlt_{op}$ or WT-associated animals were treated daily with the purified cell envelope components (WTA, PG, and LTA) for 5 d. On day 6, larvae were harvested and their size measured (*Figure 6b–e* and *Figure 6—figure supplement 1a–d*). Daily supplementation with purified WTA or PG either from WT or $\Delta dltXABCD$ strains did not impact the growth of GF or $\Delta dlt_{op}$-associated larvae (*Figure 6b and c* and *Figure 6—figure supplement 1a and b*). In contrast, the daily supplementation of purified LTA isolated from WT (D-Ala-LTAs) to larvae associated with the $\Delta dlt_{op}$ strain shows a growth-promoting effect that leads to increased larval size in this condition compared to the non-supplemented control (*Figure 6d and e*). This growth-promoting effect was not observed in GF and WT-associated animals nor when $\Delta dlt_{op}$ associated larvae were supplemented with non-alanylated-LTAs isolated from $\Delta dltXABCD$ strain (*Figure 6d and e* and *Figure 6—figure supplement 1c and d*). We repeated the similar experiment with $\Delta ltaS$-associated larvae and again observed an improved larval growth upon supplementation with LTA purified from WT (D-Ala-LTAs) but not from LTAs purified from $\Delta dltXABCD$ strain (i.e. non-alanylated-LTAs) (*Figure 6e* and *Figure 6—figure supplement 1d*). These results therefore establish that D-Ala-LTAs are necessary and sufficient to restore *Drosophila* juvenile growth.

Finally, we repeated purified LTA supplementations on *Dredd* larvae associated with the $\Delta ltaS$ strain for 5 d and dissected the midguts of size-matched larvae to analyze intestinal peptidase (*Jon66cii* and *Jon65Ai*) expression by RT-qPCR independently of any mDAP-PG sensing/Imd signaling input (*Figure 6a, f and g* and *Figure 6—figure supplement 2*). We observed that supplementation with D-Ala-LTAs was sufficient to recapitulate the effect of WT strain on intestinal peptidase expression even in the absence of any mDAP-PG signal (*Figure 5b*, *Figure 6f and g*, and *Figure 6—figure supplement 2*). Taken collectively, our results demonstrate that in addition to and independently of

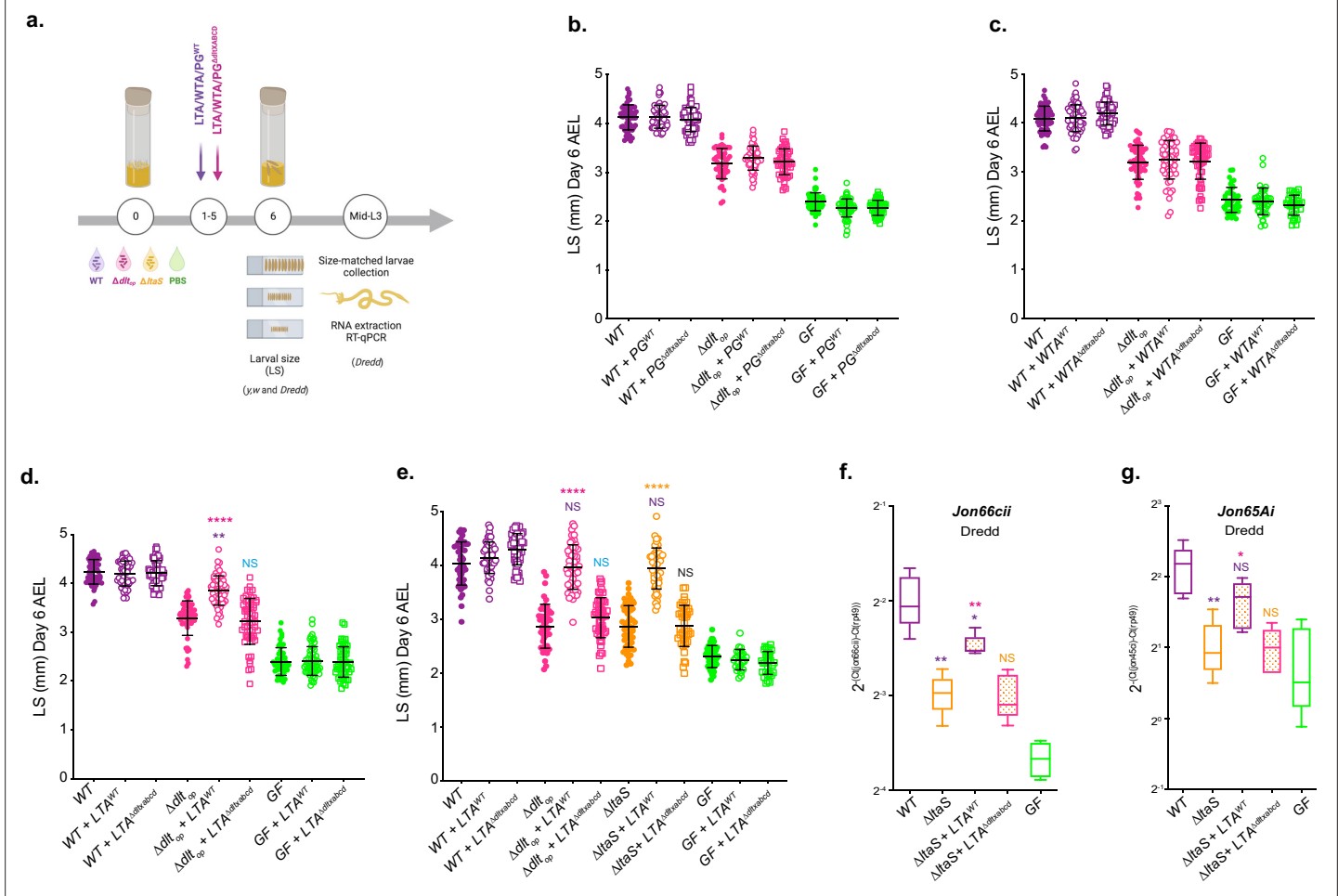

**Figure 6.** D-Ala lipoteichoic acids (LTAs) are bacterial cues supporting *Drosophila* intestinal response and juvenile growth. (**a**) Experimental set-up to test the impact of cell envelope components on *Drosophila* growth and proteases expression: germ-free (GF) eggs were inoculated with WT, Δ*dlt*$_{op}$, Δ*ltaS,* or PBS for the GF condition and supplemented daily (for 5 d) with LTA, wall teichoic acid (WTA), and peptidoglycan (PG) extracted from WT or Δ*dltXABCD* strains. 1 μg of each component purified from WT was used. For comparison with components extracted Δ*dltXABCD* strain, the final PG, WTA, and LTA suspensions were adjusted to the same amount of Gro, Rbo, and Mur, respectively. At day 6 after inoculation, larvae were harvested and measured (see the 'Materials and methods' section for details). Mid-L3 sized-matched larvae were collected for each condition. Their guts were dissected followed by RNA extraction and RT-qPCR targeting proteases expression. (**b–d**) Larval longitudinal length after inoculation with strains WT, Δ*dlt*$_{op}$, PBS, and purified PG (PG) (**b**) WTA (**c**) or LTA (**d**) from WT or Δ*dltXABCD*. Larvae were collected 6 d after the first association and measured as described in the 'Materials and methods' section. The pink asterisks represent a statistically significant difference compared with Δ*dlt*$_{op}$ larval size; purple asterisks represent a statistically significant difference compared with WT larval size. NS represents the absence of a statistically significant difference compared to Δ*dlt*$_{op}$. ****p<0.0001; **0.001<p<0.01. The bars in the graph represent means and SD. A representative graph from one out of three independent experiments is shown. (**e**) Larval longitudinal length after inoculation with strains WT, Δ*dlt*$_{op}$, Δ*ltaS,* or PBS and purified LTA from WT or Δ*dltXABCD* strains. Larvae were collected 6 d after the first association and measured as described in the 'Materials and methods' section. The pink asterisks represent a statistically significant difference compared with Δ*dlt*$_{op}$ larval size; purple asterisks represent a statistically significant difference compared with WT larval size ****p<0.0001; **0.001<p<0.01. The bars in the graph represent means and SD. A representative graph from one out of three independent experiments is shown. (**f, g**) , Mean ± SD of 2$^{-ΔCtgene/ΔCtrp49}$ ratios for *Jon66Cii* and *Jon65Ai* detected in dissected guts of *y,w,Dredd* size-matched larvae, associated with *Lp*$^{NC8}$, Δ*dlt*$_{op}$, Δ*ltaS* or the GF and LTA from WT or Δ*dltXABCD* strains, from five biological replicates. Representative graphs from one out of three independent experiments are shown. The purple asterisks represent a statistically significant difference compared with *Lp*$^{NC8}$ protease expression. The pink asterisks represent a statistically significant difference compared with Δ*ltaS* supplemented with LTA from Δ*dltXABCD* strain. NS represents the absence of a statistically significant difference compared to Δ*ltaS* condition. **0.001<p<0.01; *p<0.05.

The online version of this article includes the following source data and figure supplement(s) for figure 6:

**Source data 1.** D-Ala lipoteichoic acids (LTAs) are bacterial cues supporting *Drosophila* intestinal response and juvenile growth.

**Source data 2.** Experimental replicates from *Figure 6b–d*.

**Source data 3.** Experimental replicates from *Figure 6f and g*.

*Figure 6 continued on next page*

*Figure 6 continued*

**Figure supplement 1.** Experimental replicates from *Figure 6b-d*.

**Figure supplement 2.** Experimental replicates from *Figure 6f and g*.

mDAP-PG signaling, D-Ala-LTAs are important direct bacterial cues supporting intestinal peptidase expression and *Drosophila* juvenile growth.

## Discussion

The results obtained in this study pinpoint the bacterial machinery and molecular determinant of the bacterial cell envelope involved in the beneficial relationship between *Drosophila* and selected strains of its commensal partner *Lp,* which results in optimized host growth. Previously, we identified through forward genetic screening the *pbpX2-dltXABCD* operon encoding a bacterial machinery essential to $Lp^{NC8}$-mediated support to *Drosophila* larval growth. Here, using complementary structural, biochemical, and genetic approaches, we reveal that the protein encoded by the first gene of this operon (now renamed DltE) is not a *bona fide* PBP as it does not bind penicillin nor modifies *Lp* PG. Instead, we establish that DltE is a D-alanine esterase acting upon LTA. While studying its physiological function, we discovered that DltE contributes together with the other proteins encoded by the *dlt* operon (the DltX-A-B-C-D machinery) to the alanylation of LTA on its glycerol residues. The study of DltE's substrates gave insights on the structure and composition of $Lp^{NC8}$'s TAs, which revealed that only LTA and not WTA are alanylated in this strain. Finally, prompted by this strain-specific feature and the importance of the *dltEXABCD* operon for $Lp^{NC8}$-mediated support to *Drosophila* growth, we identified that LTAs alanylated by the DltE/X/A/B/C/D complex (but not WTA, which are not alanylated, nor non-alanylated LTA) act as direct bacterial clues supporting *Drosophila* intestinal function and juvenile growth.

### DltE and the Dlt machinery

Regardless of the initial DltE in silico annotation as a PBP (PbpX2), our structural data revealed that the global architecture of its active site cavity is reshaped and different from that of canonical PBP enzymes suggesting an alternative substrate recognition mode and/or a different substrate specificity. Hence, we show that D-Ala LTA and not PG is the substrate of DltE. Given its atomic structure and D-Ala esterase activity, DltE may rather relate to *Staphylococcus aureus* FmtA, a D-amino esterase acting on teichoic acids (*Rahman et al., 2016*). FmtA is, however, different from DltE given that it shows residual PBP activity, is able to interact with ß-lactams, and is encoded apart from the *dltABCD* operon in *S. aureus* genome. So far, the hypothesis advanced for FmtA biological function concerns the removal of D-Ala from LTAs to make it available to transfer onto WTAs or removal of D-Ala from WTA to reset cell surface charge under particular conditions (*Rahman et al., 2016*). Considering that WTA are not alanylated in $Lp^{NC8}$, these observations suggest that DltE possesses a distinct function than FmtA. We notice an apparent discrepancy between DltE enzymatic activity and the D-alanine content of the *dltE* mutants (Δ*dltE* and *dltE^S128A^*). We posit that DltE may be involved in the control of D-Ala distribution on the LTA chains and DltE inactivation would result in impaired functioning of the whole D-alanylation machinery. Our work paves the way to the characterization of the complete mechanistic framework in which the interplay between the Dlt machinery and DltE are required for the D-alanylation of LTAs.

### Strain specificity in *L. plantarum* teichoic acids composition and modifications

We analyzed $Lp^{NC8}$ TAs composition through multidimensional NMR spectroscopy and determined their alanylation levels. $Lp^{NC8}$ LTAs are, as expected from this strain's genomic information and previous Gram-positive bacteria LTA characterization (*Percy and Gründling, 2014*; *Sutcliffe, 2011*), constituted by poly-(Gro-phosphate) chains substituted, among others, with Ala. In contrast, but in agreement with the genetic information from $Lp^{NC8}$ genome, this strain produces poly(Rbo-phosphate) WTAs given the lack of the *tag* genes encoding the machinery responsible for the synthesis of poly-(Gro-phosphate) chains in its genome (*Bron et al., 2012*). The ability to produce

poly-(Gro-phosphate) WTA chains is a strain-specific feature in *Lp* as the *tag* gene cluster is only found in 22 out of 54 strains recently studied at the genomic level (*Martino et al., 2016*). However, in the case of the *Lp^WCFS1* strain, despite its genome carrying the genetic information to synthetize both types of WTA (poly-(Gro-phosphate) and poly-(Rbo-phosphate)), only poly(Gro-phosphate) WTAs were detected at steady state (*Bron et al., 2012*). Surprisingly, no Ala substituents were detected in *Lp^NC8* WTAs. This is an original feature as in other *L. plantarum* strains such as *Lp^WCFS1*, WTA appears to be D-alanylated (*Bron et al., 2012*). Yet, these WTAs are composed of poly(Gro-phosphate) chains so it would be interesting to investigate the levels of D-alanylation in other *Lp* strains harboring only poly(Rbo-phosphate) WTAs. Taken together, these observations call for future systematic analysis of TA composition and modification among *L. plantarum* strains and question the potential correlation between WTA composition and/or TA D-alanylation patterns and the ability of the strain to benefit *Drosophila* growth.

### *Drosophila* response to commensals' cell envelope determinants

We have previously established that PG fragments (*Erkosar et al., 2015*) and D-Ala esterification to the cell envelope (*Matos et al., 2017*) trigger intestinal peptidase expression and support *Drosophila* juvenile growth. Here, we report that in addition to PG fragments, D-Ala-LTA from *Lp^NC8* cell envelop represent a bacterial cue necessary and sufficient for the ability of *Lp^NC8* to promote *Drosophila* growth and upregulation of intestinal digestive peptidases in enterocytes. Given that *L. plantarum* is a gut luminal microbe and that the *Drosophila* midgut is lined with both thick chitinous matrix and thin mucus layer, yet permeable to large macromolecules and small nutrient particles (*Lemaitre and Miguel-Aliaga, 2013*), how exactly these macromolecules associated to the bacterial cell membrane reach the enterocytes remains elusive. Similarly to PG fragments, which are shaded in the extracellular media during bacterial cell division, LTA chains may also be shed after cleavage of the lipid anchor (*Neuhaus and Baddiley, 2003*). In addition, our recent findings report *Lp^NC8*'s ability to release microvesicles of multi-nanometers ranges (*Grenier et al., 2021*). This observation is coupled to previous reports showing that other *Lactobacillus* strains form microvesicles containing LTAs that are endocytosed by mammalian enterocytes (*Champagne-Jorgensen et al., 2021*), it is tempting to speculate that *Lp^NC8* D-Ala-LTAs are released via bacterial microvesicles cargo. The *Drosophila* host would capitalize on these features by endocytosing cleaved LTA or such cargo and sense these bacterial cues through pattern recognition receptors expressed in enterocytes.

PG fragments are sensed intracellularly in the *Drosophila* midgut by PGRP-LE, which signals the Imd pathway to stimulate intestinal peptidase expression. How D-Ala-LTAs are sensed and signal in enterocytes remain elusive, but our results support the notion that another signaling cascade beyond the one engaged by PGRP-LE is leading to D-Ala-LTA-mediated intestinal peptidase induction. Taken together, it seems that *Drosophila* ensures an optimal intestinal response to bacterial cues by using additive rather than interdependent signals converging locally to intestinal peptidase expression and macroscopically to enhanced juvenile growth.

Given the strong analogies of the impact of *L. plantarum* strains on *Drosophila* and mouse growth (*Schwarzer et al., 2016*) and previous work focusing on LTA in host-Gram-positive bacteria interactions (*Champagne-Jorgensen et al., 2021*; *Hara et al., 2018*), an exciting perspective will be to test the contribution of D-Ala-LTA to *L. plantarum*-mediated host growth promotion in other symbiotic models including mammals.

## Materials and methods
### *Drosophila* diets, stocks and breeding

*Drosophila* stocks were cultured as described in *Erkosar et al., 2015*. Briefly, flies were kept at 25°C with 12/12 hr dark/light cycles on a yeast/cornmeal medium containing 50 g/L of inactivated yeast. The low-yeast diet was obtained by reducing the amount of inactivated yeast to 7 g/L. Germ-free stocks were established as described in *Erkosar et al., 2015*. Axenicity was routinely tested by plating serial dilutions of animal lysates on nutrient agar plates. *Drosophila y,w* flies were used as the reference strain in this work. The following *Drosophila* line was also used: *y,w,Dredd^F64* (*Leulier et al., 2000*).

## Bacterial strains and growth conditions

Strains used in this study are listed in *Supplementary file 4*. *Escherichia coli* strains were grown at 37°C in LB medium with agitation. *L. plantarum* strains were grown in static conditions in MRS media at 37°C, unless differently stated. Erythromycin antibiotic was used at 5 µg/mL for *L. plantarum* and 150 µg/mL for *E. coli* in the context of the deletion strains construction.

## Construction of deletion strains in *L. plantarum^NC8*

Independent markerless deletions on *ltaS*, *tagO*, *dacA1*, and *dacA2* genes of *Lp^NC8* genome were constructed through homology-based recombination with double-crossing over as described by *Matos et al., 2017*. Briefly, the 5′- and 3′-terminal regions of each region were PCR-amplified with Q5 High-Fidelity 2X Master Mix (NEB) from *Lp^NC8* chromosomal DNA. Primers contained overlapping regions with pG+host9 (*Maguin et al., 1996*) to allow for Gibson Assembly. PCR amplifications were made using the following primers: XL01/XL02 and XL03/XL04 (*ltaS*), XL05/XL06 and XL07/XL08 (*tagO*), XL09/XL10 and XL11/XL12 (*dacA1*), XL13/XL14, and XL15/XL16 (*dacA2*) listed in *Supplementary file 3*. The resulting plasmids obtained by Gibson Assembly (NEB) were transformed into *Lp^NC8* electrocompetent cells and selected at the permissive temperature (28°C) on MRS plates supplemented with 5 µg/mL of erythromycin. Overnight cultures grown under the same conditions were diluted and shifted to the non-permissive temperature (41°C) in the presence of 5 µg/mL of erythromycin to select single crossover integrants. Plasmid excision by a second recombination event was promoted by growing integrants at the permissive temperature without erythromycin. Deletions were confirmed by PCR followed by sequencing. The strain deleted for *dacA1* and *dacA2* was obtained by the sequential deletion of *dacA1* followed by *dacA2*.

## Knock-in of *DltE* catalytic dead versions in *L. plantarum^NC8*

*L. plantarum^NC8* strain carrying a modified version of the *dltE* gene (*dltE^S128A*) was built by knocking in the modified sequence on Δ*dltE* strain (*Matos et al., 2017*). *dltE*-modified sequence harboring the mutation S128A was obtained by PCR with modified primers (X19). The 5′- and 3′-terminal regions of *dltE* region together with the entire *dltE* gene were PCR-amplified with Q5 High-Fidelity 2X Master Mix (NEB) from *L. plantarum^NC8* chromosomal DNA using primers XL17/XL18 and XL19/XL20 (*Supplementary file 3*). The two fragments were assembled with pG+host9 (*Maguin et al., 1996*). The resulting plasmid was transformed into Δ*dltE* electrocompetent cells and selected at the permissive temperature (28°C) on MRS plates supplemented with 5 µg/mL of erythromycin. Overnight cultures grown under the same conditions were diluted and shifted to the non-permissive temperature (41°C) in the presence of 5 µg/mL of erythromycin to select single crossover integrants. Plasmid excision by a second recombination event was promoted by growing integrants at the permissive temperature without erythromycin. *dltE^S128A* knock-in was confirmed by PCR followed by sequencing.

## Larval size measurements

Axenic adults were put overnight in breeding cages to lay eggs on sterile low-yeast diet. Fresh axenic embryos were collected the next morning and seeded by pools of 40 in tubes containing fly food. $1 \times 10^8$ CFUs or PBS were then inoculated homogeneously on the substrate and the eggs. Fly tubes are incubated at 25°C until larvae collection. *Drosophila* larvae, 6 d after inoculation, were randomly collected and processed as described by *Erkosar et al., 2015*. Individual larval longitudinal length was quantified using ImageJ software (*Schneider et al., 2012*). For the cell envelope components supplementation experiments, 1 µg of purified LTA, WTA, and PG (extracted from strains *Lp^NC8* and Δ*dltXABCD*) resuspended in PBS were added, independently, to the fly food every day until day 5 (see *Figure 6a*). For comparison of components extracted from each of the two strains, the final LTA, WTA, and PG suspensions were adjusted to the same amount of Gro, Rbo, and Mur, respectively. At day 6, larvae were harvested and larval longitudinal length was quantified using ImageJ software (*Schneider et al., 2012*).

## Developmental timing determination

Axenic adults were placed in breeding cages overnight to lay eggs on sterile poor-yeast diet. Fresh axenic embryos were collected the next morning and seeded by pools of 40 in tubes containing fly food. A total of $1 \times 10^8$ CFUs of each strain or PBS was then inoculated homogeneously on the

substrate and the eggs and incubated at 25°C. The emergence of pupae was scored every day until all pupae had emerged. D50 (day when 50% of the pupae emerged) was determined using D50App (*Matos et al., 2017*).

## Bacterial loads analysis

To access bacterial CFU in the fly nutritional matrix, microtubes containing food and larvae were inoculated with $1 \times 10^7$ CFUs of each strain, independently. Tubes were incubated at 25°C until being processed. For bacterial load quantification, 0.75–1 mm glass microbeads and 500 µL PBS were added directly into the microtubes. Samples were homogenized with the Precellys 24 tissue homogenizer (Bertin Technologies). Lysates dilutions (in PBS) were plated on MRS agar using the Easyspiral automatic plater (Intersciences). MRS agar plates were then incubated for 24 hr at 37°C. Bacterial concentration was deduced from CFU counts on MRS agar plates using the automatic colony counter Scan1200 (Intersciences) and its counting software. For larval loads, *y,w* axenic eggs were inoculated with $1 \times 10^8$ CFUs of each strain and incubated at 25°C until collection. Size-matched larvae were harvested from the nutritive substrate and surface-sterilized with a 30 s bath in 70% ethanol under agitation and rinsed in sterile water. Pools of five larvae were deposited in 1.5 mL microtubes containing 0.75–1 mm glass microbeads and 500 µL of PBS.

## RNA extraction and RT-qPCR analysis

Axenic *y,w* and *y,w,Dredd* embryos were inoculated with $1 \times 10^8$ CFU of $Lp^{NC8}$, $\Delta dlt_{op,}$ and $\Delta ltaS$ strains independently or kept axenic. Larvae were size matched for the four conditions and harvested at mid-L3 larval stage. Alternatively, larvae were inoculated with each one of the strains mentioned above and supplemented daily with cell envelop purified components (LTA, WTA, PGN) from $Lp^{NC8}$ or $\Delta dlt_{op}$ cells. RNA extraction of five replicates of six dissected guts for each condition was performed as described by *Matos et al., 2017*. RT-qPCR was performed using gene-specific primer sets (*Supplementary file 3*) as described by *Matos et al., 2017*. Results are presented as the value of $\Delta Ct^{gene}/\Delta Ct^{rp49}$.

## Statistical analysis

Data representation and analysis was performed using GraphPad PRISM 8 software (https://www.graphpad.com). A total of 3–5 biological replicates were used for all experiments performed in this study in order to ensure representativity and statistical significance. All samples were included in the analysis. Experiments were done without blinding. Two-sided Mann–Whitney's test was applied to perform pairwise statistical analyses between conditions.

## *E. coli* plasmid construction

DNA fragments were amplified by polymerase chain reaction using *L. plantarum* cDNA as a template and oligonucleotides listed in *Supplementary file 5*. The DNA encoding the extracellular domain of wild-type DltE (DltE$_{extra}$) and the catalytic mutant DltE$_{extra}$-S128A were cloned into the NcoI and XhoI sites of the pET-28a(+) vector that expresses proteins fused to a C-terminal hexahistidine (His)6 tag (*Supplementary file 6*).

## Protein production and purification

DltE$_{extra}$ and DltE$_{extra}$-S128A were expressed in *E. coli* BL21 (DE3)-RIPL cells. Cells were grown in LB media at 37°C and induced with 0.5 mM isopropyl b-D-1-thiogalactopyranoside (IPTG) overnight at 18°C. The cells were then harvested by centrifugation and resuspended in lysis buffer (50 mM Tris, pH 7.5, 500 mM NaCl, 10% [v/v] glycerol, 1 mM dithiothreitol [DTT] 0.01 mg/ml lysozyme, 0.006 mg/mL Dnase/RNase, protease inhibitor). The resuspended cells were disrupted by sonication and centrifuged at $14,000 \times g$ for 45 min. Proteins were purified by a first step of Ni-NTA affinity chromatography with the elution buffer (50 mM Tris pH 7.5, 300 mM NaCl, 1 mM DTT, 250 mM imidazole). The eluted fractions were then concentrated and buffer exchange was performed in a centrifugal filter unit with gel filtration buffer (50 mM Tris pH7.5, 100 mM NaCl, and 1 mM DTT). The proteins were finally applied to a Superdex 200 16/600 GL size-exclusion column (GE Healthcare) and eluted with gel filtration buffer.

## Crystallization, data collection, and structure determination

Crystallization conditions were screened at 293 K using the sitting-drop vapor-diffusion method and commercial crystallization kits Crystal Screen 1 and 2, PEG/Ion 1 and 2 (Hampton Research). Further optimization screenings around conditions of initial hits were also performed. The crystallization drops (0.2 μL protein solution and 0.2 μL reservoir solution) were set up using a Mosquito crystallization robot and equilibrated against 70 μL reservoir solution. DltE$_{extra}$ and DltE$_{extra}$-S128A were concentrated to around 10 mg/mL prior to crystallization. Diffraction quality crystals grew after around 1 wk in three different conditions: (1) 26% PEG MME 5K, 0.1 M Tris pH8.5, 0.15 M LiSO$_4$ and 30% ethylene glycol (named hereafter apo condition); (2) 26% PEG 3350, 0.1 M ammonium tartrate (tartrate condition); and (3) 20% PEG 3350, 0.2 M ammonium tartrate, and 0.1 M TCEP HCl (TCEP condition). Crystals obtained in condition 2 were also soaked for 1 d in the crystallization condition containing 10 mM of LTA (prepared as described below) (LTA condition).

Crystals grown in conditions 2 and 3 were cryoprotected prior to data collection by rapid soaking in mother liquor containing 20% (v/v) glycerol and for the TA-soaked crystals 10 mM of TA. Crystals were flash-cooled in liquid nitrogen and diffraction data were collected at cryogenic temperature (100 K) on beamlines ID23-2 and ID30B at the European Synchrotron Radiation Facility (ESRF, Grenoble, France) and on beamlines PROXIMA-1 and PROXIMA-2 at SOLEIL synchrotron (Gif sur Yvette, France).

Data were processed using the XDS package (*Kabsch, 2010*). The space groups, asymmetric unit contents, and diffraction resolutions obtained for each of the crystallization conditions are presented in *Supplementary file 1*. The structures were solved by molecular replacement using Phaser implemented in PHENIX (*Adams et al., 2010*). The PDB entry 3WWX (*Arima et al., 2016*) was used as starting model to solve the DltE$_{extra}$-apo structure derived from the crystals obtained in the crystallization condition 1. The other three structures (named DltE$_{extra}$-tartrate; DltE$_{extra}$-TCEP; and DltE$_{extra}$-LTA) were solved using the DltE$_{extra}$-apo as a starting model. Structures were refined using iterative rounds of COOT (*Emsley et al., 2010*) and PHENIX (*Adams et al., 2010*) or Refmac5 of CCP4 (*Murshudov et al., 1997*). The quality of the final structure was assessed with MOLPROBITY (*Chen et al., 2010*) before deposition at the Protein Data Bank under the accession codes 8AGR(DltE$_{extra}$-apo), 8AIK(DltE$_{extra}$-tartrate), 8AJI(DltE$_{extra}$-TCEP), and 8AKH (DltE$_{extra}$-LTA). Sequence alignments and structure images were generated with PyMOL (Schrödinger, LLC), UCSF ChimeraX (*Pettersen et al., 2021*), and ESPript and ENDscript (*Robert and Gouet, 2014*). Data collection and final refinement statistics are presented in *Supplementary file 1*.

## Microscale thermophoresis assays

Protein–ligand interactions were analyzed by MST (*Jerabek-Willemsen et al., 2011*). Buffer of purified and concentrated protein samples was exchanged on a desalting PD-10 column to labeling buffer containing HEPES 25 mM pH 7.5, NaCl 300 mM, Tween 20 0.05% (w/v). Proteins were then labeled with NHS red fluorescent dye according to the instructions of the RED-NHS Monolith NT Protein Labeling kit (NanoTemper Technologies GmbH, Munchen, Germany). After a short incubation of target-partner complex, the samples were loaded into MST premium glass capillaries and eight measurements were performed (four with target alone and four with target-partner complex) at 22°C. The assays were repeated three times for each binding check experiment. Data analyses were performed using Nanotemper Analysis software provided by the manufacturer.

## Release of D-Ala from whole bacterial cells and quantification by UHPLC

D-Ala esterified to teichoic acids was detected and quantified as described by *Kovács et al., 2006*. Briefly, D-Ala was released from lyophilized whole heat-inactivated bacteria by mild alkaline hydrolysis with 0.1 N NaOH for 1 hr at 37°C. After neutralization, the extract was incubated with Marfey's reagent (1-fluoro-2,4-dinitrophenyl-5-L-alanine amide; Sigma). This reagent reacts with the optical isomers of amino acids to form diastereomeric *N*-aryl derivatives, which can be separated by HPLC. Separation of the amino acid derivatives was performed on a C$_{18}$ reversed-phase column (Zorbax Eclipse Plus C18 RRHD 2.1 × 50 mm 1.8 μm Agilent) with an Agilent UHPLC 1290 system with a linear elution gradient of acetonitrile in 20 mM sodium acetate buffer pH 5.0. The eluted compounds were detected by UV absorbance at 340 nm. Quantification was achieved by comparison with D-alanine

standards in the range of 50–2000 pmol. Mean values were obtained from three independent cultures with two injections for each.

## Detection of LTA by western blot

LTA detection was done as previously described (*Webb et al., 2009*) with some modifications. Bacterial cells from a 2 mL overnight culture in MRS were added to lysing matrix tubes containing 0.1 mm silica beads (Lysing Matrix B, MP Biomedicals) and were broken using an MP Biomedicals FastPrep 24 Homogenizer (4.5 m/s intensity for 30 s), with samples being kept on ice. After centrifugation of the suspension at 200 × $g$ for 1 min to remove glass beads, cell envelopes and cell debris were recovered by centrifugation at 20.000 × $g$ for 15 min. The pellet was resuspended in sample buffer containing 2% SDS. The sample buffer volume was normalized on the culture OD600nm. Samples were heated at 95°C for 20 min and centrifuged at 20.000 × $g$ for 5 min. Supernatants were loaded on a 15% SDS-polyacrylamide electrophoresis (SDS-PAGE) gel. LTA were detected by western blotting after transfer onto a PVDF membrane (Immobilon-P transfer membrane, Millipore) by incubation with an anti-LTA mouse monoclonal antibody specific for poly-glycerolphosphate chains (clone 55, Origene, AM26274LE-N) at 1/1000 dilution, followed by anti-mouse antibody coupled to horseradish peroxidase (Thermo Fisher Scientific) at 1/2000. Western blot was revealed by chemiluminescence detection using Pierce ECL Western blotting substrate (Thermo Fisher Scientific) and a ChemiDoc imaging system (Bio-Rad).

## Preparation of *L. plantarum* cell walls

*L. plantarum* strains (500 mL) were grown overnight in MRS medium. Cell walls were prepared as described previously (*Matos et al., 2017*), with some modifications. pH of buffer solutions was kept ≤6.0 in the different steps to avoid potential D-Ala hydrolysis from teichoic acids. Briefly, bacteria inactivated by heat treatment were boiled in 5% SDS in 50 mM MES buffer pH5.5 for 25 min. After centrifugation for 10 min at 20.000 × $g$, the pellets were resuspended in 4% SDS in 50 mM MES buffer pH 5.5 and boiled again for 15 min. The pellet was washed six times with 10 mM MES pH 5.5 preheated at 60°C. An additional step was added consisting in shearing sacculi with glass beads. Pellets were resuspended in 1 mL of 10 mM MES pH 5.5, and the suspension was added to lysing matrix tubes containing 0.1 mm silica beads (MP Biomedicals). The cells were broken using an MP Biomedicals FastPrep 24 Homogenizer (4.5 m/s intensity for 30 s). After centrifugation of the suspension at 200 × $g$ for 1 min to remove glass beads, insoluble material containing cell walls was recovered by centrifugation 20.000 × $g$ for 15 min. The pellet was resuspended in 50 mM MES pH 6.0, further treated with pronase, trypsin, DNase, RNase, and lipase, and finally boiled in 4% SDS in 10 mM MES pH 5.5 for 15 min. The final pellet was washed four times with 10 mM MES pH 5.5 and twice with MilliQ H$_2$O to remove SDS traces. The purified cell walls (containing PG, WTA, and polysaccharides) was lyophilized and further used for WTA and PG purification.

## PG purification

Purified cell walls were treated with hydrofluoric acid (48%) for 19 hr at 0°C to remove WTA and polysaccharides. The remaining insoluble purified PG was washed twice times with 250 mM Tris-HCl pH 8.0 and four times with MilliQ H$_2$O and lyophilized.

## WTA purification

WTA were extracted from purified cell walls with TCA as described previously (*Tomita et al., 2009*) with some modifications. Briefly, lyophilized cell walls (50 mg; prepared as described above, keeping pH < 6.0 to avoid D-Ala hydrolysis from WTA) were incubated with 1 mL of 10% TCA at 4°C for 48 hr under rotation. The suspension was then centrifuged at 20.000 × $g$ for 20 min at 4°C and the supernatant was recovered. WTAs were precipitated by addition of 5 V of ethanol and incubation overnight at –20°C. The pellet was purified by resuspension in TCA 10% and precipitation with ethanol. The final pellet was washed twice with cold ethanol, and the pellet was resuspended in MilliQ H$_2$O and lyophilized. WTAs were resuspended in 100 mM ammonium acetate buffer pH 4.8 (buffer A) and purified by anion exchange chromatography on a 1 mL HiTrap Q HP column (Cytiva) equilibrated with buffer A. WTAs were eluted with a gradient from 0 to 100% buffer B (buffer A containing 1 M NaCl) in 20 min. Fractions were collected and the presence of WTA was assessed in microplates with

thymol-$H_2SO_4$ reagent (*Engelhardt and Ohs, 1987*), allowing detection of hexoses substituents of WTAs. The WTA-containing fractions were pooled and dialyzed against 10 mM ammonium acetate pH 4.8 with Float-a-Lyzer G2 dialysis devices (cut-off 500–1000 Da) (Spectra/Por) and lyophilized.

## LTA purification

*L. plantarum* strains (5 L) were grown overnight in MRS. Bacteria were harvested at 4.500 × *g* for 10 min. The pellet was resuspended in 5 mL of 20 mM ammonium acetate buffer pH 4.6 (around 20 OD/mL). Bacteria were broken with a Constant Systems Ltd Basic Z Cell Disrupter (CellD) at 2000 bars. Non-broken cells were discarded by centrifugation at 3.000 × *g* for 15 min. LTA was obtained by butanol extraction allowing to keep D-Ala esterified on LTAs as described previously (*Gründling and Schneewind, 2007*; *Morath et al., 2001*). The supernatant was recovered and butanol-1 (1:1 V/V) was added in a glass container. The mixture was stirred for 30 min at room temperature. Insoluble material was discarded after centrifugation at 13.000 × *g* for 20 min in PPCO tubes (Nalgene). The liquid phase was recovered and centrifuged again; the aqueous phase (lower one) containing LTA was recovered. The sample was treated with DNAse II (Sigma D8764) at 50 µg/mL for 2 hr at 37°C to degrade contaminating DNA and lyophilized. LTA was further purified by hydrophobic interaction chromatography (HIC) with a Hi-prep Octyl FF 16/10 column with an AKTA chromatography system. LTA was dissolved in buffer A (10 mM ammonium acetate pH 4.7 containing 15% propanol-1). Elution was performed with a gradient of buffer B (10 mM ammonium acetate pH 4.7 containing 60% propanol-1) from 0 to 100% in 1 hr. Fractions were collected and the presence of LTA was tested by dot blot. Aliquots were spotted on a PVDF membrane (Immobilon-P transfer membrane, Millipore) and incubated with anti-LTA monoclonal antibody (Origene, AM26274LE-N) at 1/1000 dilution, followed by anti-mouse antibody coupled to horseradish peroxidase (Thermo Fisher) at 1/2000. Dot blot was revealed by chemiluminescence detection using Pierce ECL Western blotting substrate (Thermo Scientific) and detection with a Chemidoc system (Bio-Rad). Fractions containing LTA were pooled, dialyzed against 10 mM ammonium acetate pH 4.6 with Float-a-Lyzer G2 dialysis devices (cut-off 500–1000 Da) (Spectra/Por 500–1000 Da cut-off), and lyophilized.

## PG, WTA, and LTA composition analysis and quantification

PG was quantified by analysis of muramic acid (Mur) content after acid hydrolysis as described previously (*Matos et al., 2017*). PG (400 µg) was hydrolyzed by 6 N HCl for 16 hr at 100°C under vacuum. Mur was quantified by high-performance anion exchange chromatography coupled with pulse amperometric detection (HPAEC-PAD) with an ICS5000 system (Thermo Scientific) and a Dionex CarboPac PA-1 anion exchange column (4 × 250 mm) (Thermo Scientific) with a guard column. Quantification was made with a standard curve of pure Mur (Sigma-Aldrich) (50–1000 pmol).

WTA composition was obtained after hydrolysis with hydrofluoric acid 48% at 0°C for 48 hr. After HF evaporation, samples were further treated with 4 M trifluoroacetic acid (TFA) for 3 hr at 110°C. Composition was determined by HPAEC-PAD (ICS5000 system, Thermo Fisher Scientific) with a CarboPac PA20 column (Dionex). Composition and amount were determined by comparison with standard amounts of alditols (glycerol, ribitol) and monosaccharides glucosamine, galactosamine, glucose, galactose, rhamnose, and ribose.

LTA composition was obtained after hydrolysis by hydrofluoric acid 48% at 0°C for 48 hr. After HF evaporation, samples were treated with 4 M TFA for 3 hr at 110°C. Composition was determined by HPLC analysis with an Aminex HPX-87C column (Bio-Rad) with a Waters 2414 refractive index (RI) detector using alditols (glycerol, ribitol), monosaccharides and phosphate standards. Quantification was performed by comparison with standard amounts of the same compounds. D-Ala esterified to LTA was detected after alkaline hydrolysis using Marfey's reagent derivatization, as described above for whole cells.

For supplementation experiments in fly tests, 1 µg of each purified component (PG, WTA, and LTA) was used. For comparison of components extracted from $Lp^{NC8}$ and Δ*dltXABCD*, the final PG, WTA, and LTA suspensions were adjusted to the same amount of Mur, Rbo, and Gro, respectively.

## NMR analysis

LTA was purified after butanol extraction allowing to keep D-Ala esterified on LTAs as described above. WTA was purified after TCA extraction from cell walls as described above. pH was kept below

6.0 at all purification steps of both polymers to avoid hydrolysis of the ester bond between D-Ala and the teichoic acid backbone chains. Samples were solubilized in highly enriched deuterated water (99.96% deuterium; EurisoTop, St-Aubin, France) and lyophilized. This process was repeated twice. Data were recorded on a 9.4 T spectrometer and a 18.8 T spectrometer were $^1$H resonated at 400.33 and 800.12 MHz, and $^{13}$C resonated at 100.2 and 200.3 MHz, respectively. All samples were inserted in 3 mm tubes with matching amounts of $D_2O$. Acetone was added as an internal standard, starting from a solution of 2.5 µL of acetone–10 mL of $D_2O$. All pulse sequences were taken from the Bruker library of pulse programs and then optimized for each sample. Spectral widths were 12 and 200 ppm for the proton and carbon observations, respectively. TOCSY was performed with various mixing times of 40–120 ms. Edited $^1$H-$^{13}$C HSQC spectra were recorded with 1536 data points for detection and 256 data points for indirect direction.

## Test of carboxyesterase activity on purified muropeptide

Disaccharide-depsipentapeptide (GlcNAc-MurNAc-L-Ala-D-Gln-mDAP-L-Ala-L-Lac) was purified from PG of *dacA1dacA2* double mutant of *L. plantarum*[NC8]. PG was digested with mutanolysin (Sigma-Aldrich), and the resulting soluble muropeptides were reduced by $NaBH_4$ as described previously (*Bernard et al., 2011*). The reduced muropeptides were separated by reverse-phase ultra-high-pressure liquid chromatography (RP-UHPLC) with a 1290 chromatography system (Agilent Technologies) and a Zorbax RRHD Eclipse Plus C18 column (100 by 2.1 mm; particle size, 1.8 µm; Agilent Technologies) at 50°C using ammonium phosphate buffer and methanol linear gradient. The eluted muropeptides were detected by UV absorbance at 202 nm. The peak corresponding to the disaccharide-depsipentapeptide was collected. Purified muropeptide was then incubated with 10 µg of purified DltE$_{extra}$ or DacA1 in 50 mM Tris-HCl, 100 mM NaCl buffer overnight at 37°C. The reaction mixtures were analyzed by RP-UHPLC as described above and by LC-MS with an UHPLC instrument (Vanquish Flex, Thermo Scientific) connected to a Q Exactive Focus mass spectrometer (Thermo Scientific). Mass spectra were collected over the range m/z = 380–1400. Data were processed using Xcalibur QualBrowser v2.0 (Thermo Scientific).

## Test of D-Ala esterase activity on LTA

Purified DltE$_{extra}$ (100 and 200 µg), DltE$^{S128A}$$_{extra}$ (100 µg), or DacA1 (100 µg) were incubated overnight at 37°C with 200 µg D-Ala-LTA purified from *L. plantarum*[NC8] in a final volume of 100 µL of 20 mM MES, 150 mM NaCl pH 6.0. As a control, 200 µg of D-Ala-LTA was incubated in the same conditions without enzyme. After lyophilization, samples were treated with 48% hydrofluoric acid for 48 hr at 0°C, allowing LTA depolymerization without release of D-Ala esterified on Gro residues. Gro-D-Ala subunits and free D-Ala released from LTA chains were analyzed by LC-MS using an UHPLC instrument (Vanquish Flex, Thermo Scientific) connected to a Q Exactive Focus mass spectrometer (Thermo Fisher Scientific). A reverse-phase column (Hypersyl Gold AQ C18 column; 200 by 2.1 mm; particle size 1.9 µm; Thermo Fisher Scientific) was used for separation. To enhance the retention and resolution of the column, heptafluorobutyric acid (HFBA) was used as an ion-pairing agent. Samples were diluted 20-fold in buffer A with 2 µL injected onto the column. Buffer A contained 0.2% HBFA in MilliQ $H_2O$ and buffer B contained 0.2% HBFA in acetonitrile/MilliQ $H_2O$ (80:20, V/V). Elution was performed at flow rate 0.3 mL/min using an isocratic step of buffer A for 3 min followed by a gradient to 5% of buffer B in 5 min. Mass analysis was performed in single ion monitoring (SIM) mode by following Ala and Ala-Gro with the respective m/z of 90.05 and 164.09. Ala and Gro-Ala were identified by their MS and MS$^2$ spectra.

## Acknowledgements

The authors thank Octobre Clocher, Dali Ma, and Cathy Ramos for their contribution to the early stages of the bacteriology and *Drosophila* tasks of this project, Céline Freton for help in microscale thermophoresis, Laurane Bernelin for her contribution at the initial stages of the DltE production and crystallization, and Emmanuel Maes and Jessica L Davis for technical assistance. We acknowledge the SFR Biosciences (UAR3444/US8) for access to the Protein Science (crystallogenesis robots) and the ArthroTools platform (*Drosophila* facility) and support on the beamlines ID23-2 and ID30B at the European Synchrotron Radiation Facility (ESRF, Grenoble, France) and on PROXIMA-1 and PROXIMA-2A at SOLEIL synchrotron (Gif sur Yvette, France), the PAGes core facility (http://plateforme-pages.

univ-lille1.fr) and UAR 2014 /US 41 (Plateformes Lilloises en Biologie & Santé) for providing the scientific and technical environment conducive to achieve this work. We acknowledge ChemSyBio (Micalis, INRAE, France) for access to mass spectrometry facilities. This work was funded by the collaborative grant ANR-18-CE15-0011 to FL, MPCC and CG. Work in the FL lab was also supported by a FRM grant (DEQ20180839196), a FINOVI starting grant to RCM and an ENSL Emergence grant to RCM. Work in the CG lab was also supported by the Fondation Bettencourt-Schueller.

## Additional information

### Funding

| Funder | Grant reference number | Author |
| --- | --- | --- |
| Agence Nationale de la Recherche | ANR-18-CE15-0011 | François Leulier Marie-Pierre Chapot-Chartier Christophe Grangeasse |
| Fondation pour la Recherche Médicale | DEQ20180839196 | François Leulier |
| Fondation Innovations en Infectiologie | | Renata C Matos |
| ENS de Lyon | Emergence | Renata C Matos |
| Fondation Bettencourt Schueller | | Christophe Grangeasse |

The funders had no role in study design, data collection and interpretation, or the decision to submit the work for publication.

### Author contributions

Nikos Nikolopoulos, Formal analysis, Investigation, Methodology, Writing - review and editing; Renata C Matos, Stephanie Ravaud, Conceptualization, Formal analysis, Supervision, Investigation, Methodology, Writing – original draft; Pascal Courtin, Conceptualization, Formal analysis, Supervision, Investigation, Methodology; Houssam Akherraz, Simon Palussiere, Virginie Gueguen-Chaignon, Marie Salomon-Mallet, Alain Guillot, Investigation, Methodology; Yann Guerardel, Conceptualization, Formal analysis, Supervision, Validation, Investigation, Methodology, Writing - review and editing; Marie-Pierre Chapot-Chartier, Conceptualization, Formal analysis, Supervision, Funding acquisition, Validation, Writing – original draft, Project administration; Christophe Grangeasse, Conceptualization, Formal analysis, Supervision, Funding acquisition, Validation, Project administration, Writing - review and editing; François Leulier, Conceptualization, Formal analysis, Supervision, Funding acquisition, Validation, Writing – original draft, Project administration, Writing - review and editing

### Author ORCIDs

Renata C Matos ⓘ http://orcid.org/0000-0001-7480-6099
Stephanie Ravaud ⓘ http://orcid.org/0000-0001-5867-0785
Marie-Pierre Chapot-Chartier ⓘ http://orcid.org/0000-0002-4947-0519
François Leulier ⓘ http://orcid.org/0000-0002-4542-3053

### Decision letter and Author response

Decision letter https://doi.org/10.7554/eLife.84669.sa1
Author response https://doi.org/10.7554/eLife.84669.sa2

## Additional files

### Supplementary files

• Supplementary file 1. Data collection and refinement statistics.

• Supplementary file 2. Proton and carbon chemical shifts of Glc, Gro, and Ala constituents of lipoteichoic acid (LTA) purified from *L. plantarum* WT.

- Supplementary file 3. Primers used in this study for *Lp* genetic manipulation.
- Supplementary file 4. Bacterial strains and plasmids used in this study.
- Supplementary file 5. Primers used for *E. coli* plasmid constructions.
- Supplementary file 6. Plasmids used in this study for protein expression.
- MDAR checklist

## Data availability

Diffraction data have been deposited in PDB under the accession codes 8AGR/8AIK/8AJI/8AKH-All data generated during this study are included in the manuscript and supporting file and a single Source Data file has been provided.

The following datasets were generated:

| Author(s) | Year | Dataset title | Dataset URL | Database and Identifier |
|---|---|---|---|---|
| Nikopoulos N, Ravaud S, Grangeasse C | 2022 | Crystal structure of DltE from L.plantarum, apo form | https://www.rcsb.org/structure/8AGR | RCSB Protein Data Bank, 8AGR |
| Nikopoulos N, Ravaud S, Grangeasse C | 2022 | Crystal structure of DltE from L.plantarum, tartare bound form | https://www.rcsb.org/structure/8AIK | RCSB Protein Data Bank, 8AIK |
| Nikopoulos N, Ravaud S, Grangeasse C | 2022 | Crystal structure of DltE from L.plantarum, TCEP bound form | https://www.rcsb.org/structure/8AJI | RCSB Protein Data Bank, 8AJI |
| Nikopoulos N, Ravaud S, Grangeasse C | 2022 | Crystal structure of DltE from L.plantarum soaked with LTA | https://www.rcsb.org/structure/8AKH | RCSB Protein Data Bank, 8AKH |

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
