## [Editor Report]

This is an important study on the role of a bacterial cell wall component, D-alanylated lipoteichoic acid, as a cue in *Drosophila melanogaster*-microbiome interactions. Overall, the evidence presented to support the conclusions is compelling. The approach combines crystallography with biochemical and cellular assays that take advantage of both fly and bacterial mutants to demonstrate a physiological role in juvenile growth promotion. The work will be of broad interest to those studying host-microbe interactions, particularly aspects related to immunology and metabolism, mediated by the microbiome.

---

## [Decision Letter]

**Decision letter after peer review:**

Thank you for submitting your article "Structure-Function analysis of Lactiplantibacillus plantarum DltE reveals D-alanylated lipoteichoic acids as direct symbiotic cues supporting *Drosophila* juvenile growth" for consideration by *eLife*. Your article has been reviewed by 3 peer reviewers, one of whom is a member of our Board of Reviewing Editors, and the evaluation has been overseen by Bavesh Kana as the Senior Editor. The reviewers have opted to remain anonymous.

Essential revisions:

1) The reviewers find the use of the term "direct symbiotic cue" to not be fully supported by the data.

(1A) At this stage of the work it is not clear how widespread the release of D-Ala-LTA is. It is not clear if this is a response related strictly to symbionts, therefore the reviewers recommend that D-Ala-LTA is defined as "bacterial cue".

(1B) The reviewers also thought that the current results are not sufficient to differentiate if D-Ala-LTA is a nutritional response or cue. We reason that additional experiments to clarify this point would make the paper stronger. We explain our reasoning here and suggest that the authors either perform the suggested experiment or change the wording while discussing the caveats of the present results.

Regarding the concerns, specifically, reviewer #2 is concerned with the lower loads of the dltE mutant in Figure 4 (and the lack of data regarding the loads in Figure 6). The effect on the larvae size and development could be a result of a possible growth defect of the mutant. A possible experiment to address this point would be one with 10X more mutants added to the food to determine if the phenotype is still the same or not. In addition, the same rationale could be applied to Figure 6, where the effect of adding 10X more ltaS mutant could be tested and compared with adding 1X mutant + 10X LTA. This would enable us to determine if the addition of more mutants or not rescues the phenotype if the response is nutritional. Moreover, if LTA is a cue and not a nutrition source adding 1X LTA or 10X LTA should not make a difference. It could be that the 1X LTA used in the experiment is already a large amount so perhaps adding less than 1X and determining if there is a lower effect on larvae size (or development) is another possibility. We acknowledge that differentiating between nutritional response and bacteria cue might be difficult, therefore we leave the option to the authors to perform these experiments or to change the wording discussing this issue and acknowledging the caveat, and adapted the text to the potential new or the current results.

2) Please respond to the question raised by reviewer 1 related to the role of DltE esterase activity and the phenotype of the corresponding mutant.

3) Data presented and written text. In some data figures, only a representative experiment is shown, all the experiments should be shown (replicate experiments can be shown in the supplement). Confirm the NMR results shown in Figure 3 and add any missing data. Regarding abbreviations, please confirm that all abbreviations and acronyms are spelled out, reduce the use of abbreviations to the minimum needed, and correct formatting errors.

*Reviewer #1 (Recommendations for the authors):*

1. The results presented clearly show that DltE is an esterase that leads to the release of free D-Ala from D-alanylated-LTA. However, the results also show that dltE mutants have lower D-Ala-LTA, shown by the lower release of D-Ala from alkaline hydrolysis from whole cells of dltE mutants (Figure 4a). These results were interpreted as an indication that the esterase activity of DltE contributes to the D-alanylation of LTA, I do not understand how this is possible and did not see a possible explanation/discussion for this interpretation in the manuscript. Could it be that dltE mutants have a polar effect in the rest of the operon and that is the explanation for the lower D-alanylation of LTA in this mutant? This could also affect the interpretation of the results of panels 4b and 4c. Please clarify/discuss this apparent contradiction.

2. NMR results shown in Figure 3b need to be clarified. Legend of Figure 3b says that "Multidimensional NMR analysis of LTA isolated from WT L. plantarum" are shown but in the text, it is written that analysis of LTA from WT was compared with that from LTA extracted from the dlt operon mutant to confirm the absence of Ala residues in the mutant. This is an important result of the study and these NMR results need to be shown.

*Reviewer #2 (Recommendations for the authors):*

Overall, I find the work compelling. My concerns are listed below.

1. I find the use of the term symbiosis to be misleading. Symbiosis suggests an intimate association between a specific bacterium and the host, as in the symbiosis between the bobtail squid and Vibrio fischeri. The squid in that case makes the exquisite selection of its partner bacterial strain. As D-Alanylated lipoteichoic acids are not unique to Lactiplantibacillus plantarum or even to lactic acid bacteria in general (it is quite generic), it is misleading to call D-Ala LTA a symbiosis factor. Larvae similarly need RNA to grow, but we do not call it a symbiotic cue. Further mechanistic work to understand how larvae use D-Ala LTA is needed to ascertain whether or not there is any symbiotic cue provided by this specific molecule. To clarify my comment, I would find a title such as, "Structure-Function analysis of Lactiplantibacillus plantarum DltE reveals D-alanylated lipoteichoic acids as direct cues supporting *Drosophila* juvenile growth" to be appropriate.

2. line 258-263 and Figure 4. There is a ~10-fold decrease in CFUs/mL in the mutants at Day 5-7, which makes sense given the cell envelope could be weakened in the mutant. This reduction in CFUs corresponds to a period of rapid growth for the larvae and the reduction in bacterial numbers presumably leads to a proportional decrease in total bacterial metabolism of the food and/or in bacterial bulk nutrition as supplied in proteins and fats etc. Is it known what decrease in growth should be expected? Does the metabolomic composition of the food differ between the bacterial mutants at days 5-7? The concern could be addressed by supplementing the food with 10x more of the mutants to see if this rescues the growth defect. These points should be considered in the text to differentiate the effects of bulk nutrition from growth promotion by a specific symbiotic metabolite.

3. Figure 5a. Why is only a representative replicate shown? What factors affect the reproducibility of absolute numbers? Could the replicates be combined after some normalization? The other replicates should at least be shown in the supplement.

Regarding Figure 6, what is the difference between panels D and E besides the addition of the ∆ltaS mutant?

Panels F and G: why is only one representative replicate shown? The data are normalized, so why are they not combined? Were statistics calculated for the aggregate of the 5 biological replicates or for the one representative replicate? Were there 3 or 5 biologial repliactes. The figure legend is confusing. All the replicates should be shown in the supplement if they cannot be combined in a single panel.

4. Figure 6d,e. Similar to point 3, all replicates need to be shown somewhere, somehow in the manuscript.

5. What are the CFUs in Figure 6d,e? This is a beautiful experiment that is critical to the findings. However, one question is whether the addition of D-Ala LTA is supporting increased bacterial growth and thus D-Ala LTA is an indirect nutritional cue. This draws the necessary and sufficient claim into question as well as the claim in the title that the role of D-Ala LTA is direct. Similar to Figure 4, quantifying the CFUs in the experiment would address this concern.

6. I am not a structural biologist, so I have not done an in-depth technical review of the structural data or arguments. This includes Figures 1-3 and the corresponding supplemental figures. Assuming the conclusions drawn from these data are correct, then I think the paper makes a nice advance in understanding the molecular biology of the cues produced by L. plantarum and sensed by *Drosophila*. The work builds upon a strong foundation of previous studies from the Leulier lab and makes a significant advance.

To conclude, I think the work is strong and makes a solid advance. My concerns come down to just two words, "symbiotic" and "direct". I think the authors could remedy these concerns by modifying the writing and/or by performing additional experiments.

*Reviewer #3 (Recommendations for the authors):*

Overall, the manuscript was interesting to reach and the data compelling. My only suggestions are in the writing- it would improve the readability and breadth of the manuscript to reduce the jargon and properly introduce words before referring to them as acronyms. Given the work intersects multiple fields it would help to better define or refrain from use unless absolutely necessary to make the text more readily understood by a broader audience. Related, the figures often leave too much up to the interpretation of the reader and more specific labeling of axes would help (for example that figure 4 is referring to larval length or time to pupation is not obvious and could be more explicit).

Specific comments:

Abstract – LTAs WTAs used by not defined.

Intro-

L68 & 71 – Ap and Lp are used for microbiome members, but not defined.

L73: should be "This recognition signal….promotes".

L345: envelope is missing an 'e'.

L405&406: should be 'shed' not 'shaded'.

Figures:

Figure 4 – use clearer axis titles (larval length, time to pupation….). Also, 4d – not really an evolution experiment, so the legend is odd – would make more sense to say microbe abundance/CFUs over time or such (maybe just cut text to 'Number of CFUs on fly food and in larvae….). For 4d it is also very difficult to follow the different symbols per day, they are quite small and at this quality blurry – maybe use headings too?

Figure 5 and 6 – same comment about axis titles as above. Also, higher quality images are needed – perhaps increase symbols/lines in size or weight.

---

## [Author Response]

Essential revisions:1) The reviewers find the use of the term "direct symbiotic cue" to not be fully supported by the data.

We have agree with this statement and have removed “symbiotic” in our title.

(1A) At this stage of the work it is not clear how widespread the release of D-Ala-LTA is. It is not clear if this is a response related strictly to symbionts, therefore the reviewers recommend that D-Ala-LTA is defined as "bacterial cue".

We agree with this statement. We have substituted ‘symbiotic cues’ by ‘bacterial cues’ all along our text.

(1B) The reviewers also thought that the current results are not sufficient to differentiate if D-Ala-LTA is a nutritional response or cue. We reason that additional experiments to clarify this point would make the paper stronger. We explain our reasoning here and suggest that the authors either perform the suggested experiment or change the wording while discussing the caveats of the present results.Regarding the concerns, specifically, reviewer #2 is concerned with the lower loads of the dltE mutant in Figure 4 (and the lack of data regarding the loads in Figure 6). The effect on the larvae size and development could be a result of a possible growth defect of the mutant. A possible experiment to address this point would be one with 10X more mutants added to the food to determine if the phenotype is still the same or not. In addition, the same rationale could be applied to Figure 6, where the effect of adding 10X more ltaS mutant could be tested and compared with adding 1X mutant + 10X LTA. This would enable us to determine if the addition of more mutants or not rescues the phenotype if the response is nutritional. Moreover, if LTA is a cue and not a nutrition source adding 1X LTA or 10X LTA should not make a difference. It could be that the 1X LTA used in the experiment is already a large amount so perhaps adding less than 1X and determining if there is a lower effect on larvae size (or development) is another possibility. We acknowledge that differentiating between nutritional response and bacteria cue might be difficult, therefore we leave the option to the authors to perform these experiments or to change the wording discussing this issue and acknowledging the caveat, and adapted the text to the potential new or the current results.

As suggested by the reviewers we supplemented the fly food with 10x more bacteria to test for the response of *Drosophila* to increased amounts of bacteria. Adding 10x more WT and mutant leads to a slight increase on larval size for both strains when compared to the 1x dose (Author response image 1). The increase is common to both bacterial conditions. Of note, 1x WT is still more beneficial than adding 10x more mutant. This effect is reproduced at the level of the time to pupation (Author response image 1). Both experiments demonstrate that the effect on growth of *Lp* is not a mere nutritional effect.

Regarding the association of the bacteria with *Drosophila* larvae, we inoculate 10x more bacteria and followed their persistence in the fly food. At both concentrations, there were no significative differences between the two strains at the different time points (Author response image 1).

We then repeated the experiment from Figure 6e with a higher concentration of purified LTA, thus we added 5x more LTA from WT or ∆*dltXABCD* strains. Results shown in Figure R1d clearly shows that adding more LTA doesn’t improve the growth promoting phenotype of ∆*dlt_op_* and ∆*ltaS* when compared to the 1x LTA condition. This demonstrates again that d-Ala-LTAs are not a nutritional source but rather a signaling cue and this is the d-alanylation of LTA that act as a cue not the actual amount of LTA molecule.

**Author response image 1. sa2fig1:** (a) Larval size (corresponding to larval longitudinal length at day 6 after egg laying) (b) time to pupation (D50, corresponding to the day in which 50% of the larvae pupate) of eggs associated with different concentrations of WT and D*ltaS* strains. 1x corresponds to 10^8^ CFUs while 10x corresponds to 10^9^ CFUs. (c) Number of CFUs on fly food and larvae at days 3, 5, and 7 after inoculation with WT and ∆*ltaS* at different concentrations (D0) of L1 larvae: 1x corresponds to 10^8^ CFUs while 10x corresponds to 10^9^ CFUs. (d) Larval longitudinal length after inoculation with strains WT, ∆*dlt_op_*, ∆*ltaS* or PBS and different amounts of purified lipoteichoic acid (LTA) from WT or ∆*dltXABCD* strains. On the right panel we used 5x more purified LTA than on the left panel (Panel from the main manuscript). Larvae were collected six days after the first association and measured as described in the Methods.

2) Please respond to the question raised by reviewer 1 related to the role of DltE esterase activity and the phenotype of the corresponding mutant.

Despite the D-Ala esterase activity of DltE detected in vitro on D-alanylated LTAs, we indeed observed a substantial decrease of D-Ala esterified to LTA of D*dltE* as well as in *dltE^S148A^* mutant cells*.* In the *dltE^S148A^* catalytic mutant, the active site Ser148 is replaced by an Alanine thus the mutant and WT *dltE* genes differ only by a few nucleotides in composition. Therefore, in this latter mutant we can rule out that the mutation results in a polar effect on the rest of the operon. Because DltES148A recombinant protein is also effectively produced and stable in vitro, we can also exclude an issue with the stability of the mutated form. Regarding the deletion mutant (D*dltE)*, we kept the reading frame by keeping the 2 first and 2 last amino acids on the bacterial genome. We would thus exclude a polar effect in both cases.

To explain this apparent discrepancy between the DltE enzymatic activity and the D-Ala level in the mutants, we propose that in vivo, DltE could have transesterase activity. As a protein with a classical b-lactamase fold and a catalytic motif SxxK, DltE is expected to form a covalently bound enzyme-substrate intermediate. in vitro, in aqueous solution, hydrolysis of the acyl-enzyme intermediate leads to free D-Ala, resulting in the detected D-Ala esterase activity of DltE. in vivo, inside the cell wall, we propose that the acyl-enzyme may react with the hydroxyl group of a glycerol moiety of another LTA chain or inside the same LTA chain, leading to the transfer of D-Ala from one glycerol to another one, corresponding to a transesterase activity. DltE would thus be involved in the control of D-Ala distribution on the LTA chains and DltE inactivation could thus result in impaired functioning of the whole D-alanylation machinery. In agreement with this model we have preliminary results demonstrating direct interactions between DltE and DltD. Future work will focus on revealing the biological relevance of these interactions in the context of the Dlt machinery activity.

We have included the following sentence on the Discussion section (lane 399-401): “We posit that DltE may be involved in the control of D-Ala distribution on the LTA chains and DltE inactivation would result in impaired functioning of the whole D-alanylation machinery”.

3) Data presented and written text. In some data figures, only a representative experiment is shown, all the experiments should be shown (replicate experiments can be shown in the supplement). Confirm the NMR results shown in Figure 3 and add any missing data. Regarding abbreviations, please confirm that all abbreviations and acronyms are spelled out, reduce the use of abbreviations to the minimum needed, and correct formatting errors.

We have now added replicates to the supplementary figures together with their respective source data.

A new supplementary figure has been added (Figure 3 —figure supplement 3) showing the NMR analysis of the D*dlt_XABCD_* strain. It clearly demonstrates that the mutant LTAs are not substituted by Ala.

The readability of the text has been improved as suggested by the reviewers.